# The spatial frequency representation predicts category coding in the inferior temporal cortex

Ramin Toosi[1], Behnam Karami[2,3], Roxana Koushki[2], Farideh Shakerian[4], Jalaledin Noroozi[2], Ehsan Rezayat[5], Abdol-Hossein Vahabie[1,5], Mohammad Ali Akhaee[1]*, Mohammad-Reza A Dehaqani[1,2,4]*

[1]School of Electrical and Computer Engineering, College of Engineering, University of Tehran, Tehran, Islamic Republic of Iran; [2]School of Cognitive Sciences, Institute for Research in Fundamental Sciences, Tehran, Iran; [3]Donders Centre for Cognitive Neuroimaging, Donders Institute for Brain, Cognition, and Behaviour, Radboud University, Nijmegen, Netherlands; [4]Department of Brain and Cognitive Sciences, Cell Science Research Center, Royan Institute for Stem Cell Biology and Technology, ACECR, Tehran, Iran; [5]Department of Cognitive Sciences, Faculty of Psychology and Education, University of Tehran, Tehran, Iran

**\*For correspondence:**
akhaee@ut.ac.ir (MAA);
dehaqani@ut.ac.ir (M-RAD)

**Competing interest:** The authors declare that no competing interests exist.

## eLife Assessment

This **useful** study aimed to examine the relationship of spatial frequency selectivity of single macaque inferotemporal (IT) neurons to category selectivity. Interesting findings in this report suggest a shift in preferred spatial frequency during the response, from low to high spatial frequencies. This agrees with a coarse-to-fine processing strategy, which is in line with multiple studies in the early visual cortex. Some of the findings were difficult to evaluate because the methods are **incomplete**. The conclusion that single-unit spatial frequency selectivity can predict object coding requires further evidence to confirm.

**Abstract** Understanding the neural representation of spatial frequency (SF) in the primate cortex is vital for unraveling visual processing in object recognition. While many studies focus on SF in the primary visual cortex, the characteristics and interaction of SF with category representation remain unclear. To explore SF representation in the inferior temporal (IT) cortex of macaques, we conducted extracellular recordings with complex stimuli systematically filtered by SF. Our findings reveal explicit SF coding at both single-neuron and population levels. Temporal dynamics analysis of SF representation reveals that low SF (LSF) is decoded faster than high SF (HSF), and the SF preference dynamically shifts from LSF to HSF over time. Additionally, the SF representation for each neuron forms a profile that predicts category selectivity at the population level. IT neurons cluster into four groups based on SF preference, each with distinct category coding behaviors. Notably, HSF-preferring neurons show the highest category decoding for faces. Despite the existing connection between SF and category coding, we have identified uncorrelated representations of SF and category. Unlike category coding, SF is more sparsely represented and depends more on individual neurons. These findings dissociate SF and category representations, underscoring SF's pivotal role in object recognition.

## Introduction

Spatial frequency (SF) constitutes a pivotal component of visual stimuli encoding in the primate visual system, encompassing the number of grating cycles within a specific visual angle. Higher SF (HSF) corresponds to intricate details, while lower SF (LSF) captures broader information. Previous psychophysical studies have compellingly demonstrated the profound influence of SF manipulation on object recognition and categorization processes (*Joubert et al., 2007*; *Schyns and Oliva, 1994*; *Ashtiani et al., 2017*; *Caplette et al., 2014*; *Cheung and Bar, 2014*; *Craddock et al., 2013*). *Jahfari et al., 2013* and *Saneyoshi and Michimata, 2015* have highlighted the significance of HSF and LSF for categorical/coordinate processing and in object recognition and decision-making, respectively. The sequence in which SF content is presented also affects the categorization performance, with coarse-to-fine presentation leading to faster categorizations (*Kauffmann et al., 2015*). Considering face as a particular object, several studies showed that middle and higher SFs are more critical for face recognition (*Cheung et al., 2008*; *Costen et al., 1996*; *Fiorentini et al., 1983*; *Hayes et al., 1986*). Another vital theory suggested by psychophysical studies is the coarse-to-fine perception of visual stimuli, which states that LSF or global contents are processed faster than HSF or local contents (*Gao and Bentin, 2011*; *Kauffmann et al., 2015*; *Rokszin et al., 2016*; *Rotshtein et al., 2010*; *Schyns and Oliva, 1994*; *Yardley et al., 2012*). Despite the extensive reliance on psychophysical studies to examine the influence of SF on categorization tasks, our understanding of SF representation within primate visual systems, particularly in higher visual areas like the inferior temporal (IT) cortex, remains constrained due to the limited research in this specific domain.

One of the seminal studies investigating the neural correlates of SF processing and its significance in object recognition was conducted by *Bar, 2003*. Their research proposes a top–down mechanism driven by the rapid processing of LSF content, facilitating object recognition (*Bar, 2003*; *Fenske et al., 2006*). The exploration of SF representation has revealed the engagement of distinct brain regions in processing various SF contents (*Bastin et al., 2013*; *Bermudez et al., 2009*; *Chaumon et al., 2014*; *Cheung and Bar, 2014*; *Fintzi and Mahon, 2014*; *Gaska et al., 1988*; *Iidaka et al., 2004*; *Oram and Perrett, 1994*; *Peyrin et al., 2010*). More specifically, the orbitofrontal cortex has been identified as accessing global (LSF) and local (identity; HSF) information in the right and left hemispheres, respectively (*Fintzi and Mahon, 2014*). The V3A area exhibits low-pass tuning curves (*Gaska et al., 1988*), while HSF processing activates the left fusiform gyrus (*Iidaka et al., 2004*). Neural responses in the IT cortex, which play a pivotal role in object recognition and face perception, demonstrate correlations with the SF components of complex stimuli (*Bermudez et al., 2009*). Despite the acknowledged importance of SF as a critical characteristic influencing object recognition, a more comprehensive understanding of its representation is warranted. By unraveling the neural mechanisms underlying SF representation in the IT cortex, we can enrich our comprehension of the processing and categorization of visual information.

To address this issue, we investigate the SF representation in the IT cortex of two passive-viewing macaque monkeys. We studied the neural responses of the IT cortex to intact, SF-filtered (five ranges), and phase-scrambled stimuli. SF decoding is observed in both population- and single-level representations. Investigating the decoding pattern of individual SF bands reveals a course-to-fine manner in recall performance where LSF is decoded more accurately than HSF. Temporal dynamics analysis shows that SF coding exhibits a coarse-to-fine pattern, emphasizing faster processing of lower frequencies. Moreover, SF representation forms an average LSF-preferred tuning across neuron responses at 70–170 ms after stimulus onset. Then, the average preferred SF shifts monotonically to HSF in time after the early phase of the response, with its peak at 220 ms after the stimulus onset. The LSF-preferred tuning turns into an HSF-preferred one in the late neuron response phase.

Next, we examined the relationship between SF and category coding. We found a strong positive correlation between SF and category coding performances in sub-populations of neurons. SF coding capability of individual neurons is highly correlated with the category coding capacity of the sub-population. Moreover, clustering neurons based on their SF responses indicates a relationship between SF representation and category coding. Employing the neuron responses to five SF ranges considering only the scrambled stimuli, an SF profile was identified for each neuron that predicts the categorization performance of that neuron in a population of the neurons sharing the same profile. Neurons whose response increases with increasing SF encode faces better than other neuron populations with other profiles.

Given the co-existence of SF and category coding within the IT cortex and the prediction capability of SF for category selectively, we examined the neural mechanisms underlying SF and category representation. In single level, we found no correlation between SF and category coding capability of single neurons. At the population level, we found that the contribution of neurons to SF coding did not correlate with their contribution to category coding. Delving into the characteristics of SF coding, we found that individual neurons carry more independent SF-related information compared to the encoding of categories (face vs. non-face). Analyzing the temporal dynamics of each neuron's contribution to population-level SF coding reveals a shift in sparsity during different phases of the response. In the early phase (70–170 ms), the contribution is more sparse than category coding. However, this behavior is reversed in the late phase (170–270 ms), with SF coding showing a less sparse contribution.

Finally, we compared the representation of SF in the IT cortex with several popular convolutional neural networks (CNNs). We found that CNNs exhibited robust SF coding capabilities with significantly higher accuracies than the IT cortex. Like the IT cortex, LSF content showed higher decoding performance than the HSF content. However, while there were similarities in SF representation, CNNs did not replicate the SF-based profiles predicting neuron category selectivity observed in the IT cortex. We posit that our findings establish neural correlates pertinent to behavioral investigations into SF's role in object recognition. Additionally, our results shed light on how the IT cortex represents and utilizes SF during the object recognition process.

## Results

### SF coding in the IT cortex

To study the SF representation in the IT cortex, we designed a passive stimulus presentation task (*Figure 1a*, see Materials and methods). The task comprises two phases: the selectivity and the main. During the selectivity phase, 155 stimuli, organized into two super-ordinate and four ordinate categories, were presented (with a 50-ms stimulus presentation followed by a 500-ms blank period, see Materials and methods). Next, the six most responsive stimuli are selected along with nine fixed stimuli (six faces and three non-face objects, *Figure 1b*) to be presented during the main phase (33-ms stimulus presentation followed by a 465-ms blank, see Materials and methods). Each stimulus is phase scrambled, and then the intact and scrambled versions are filtered in five SF ranges (R1–R5, with R5 representing the highest frequency band, *Figure 1b*), resulting in a total of 180 unique stimuli presented in each session (see Materials and methods). Each session consists of 15 blocks, with each stimulus presented once per block in a random order. The IT neurons of passive-viewing monkeys are recorded where the cells cover all areas of the IT area uniformly (*Figure 1a*). We only considered the responsive neurons (see Materials and methods), totaling 266 (157 M1 and 109 M2). A sample neuron (neuron #155, M1) peristimulus time histogram is illustrated in *Figure 1c* in response to the scrambled stimuli for R1, R3, and R5. R1 exhibits the most pronounced firing rate, indicating the highest neural activity level. In contrast, R5 displays the lowest firing rate, suggesting an LSF-preferred trend in the neuron's response. To explore the SF representation and coding capability of IT neurons, each stimulus in each session block is represented by an $N$-element vector where the $i$th element is the average response of the $i$th neuron to that stimulus within a 50-ms time window (see Materials and methods).

To assess whether individual neurons encode SF-related information, we utilized the linear discriminant analysis (LDA) method to predict the SF range of the scrambled stimuli based on neuron responses (see Materials and methods). *Figure 1d* displays the average time course of SF discrimination accuracy across neurons. The accuracy value is normalized by subtracting the chance level (0.2). At single level, the accuracy surpasses the chance level by an average of 4.02% at 120 ms after stimulus onset. We only considered neurons demonstrating at least three consecutive time windows with accuracy significantly greater than the chance level, resulting in a subset of 105 neurons. The maximum accuracy of a single neuron was 19.08% higher than the chance level (unnormalized accuracy is 39.08%, neuron #193, M2). Subsequently, the SF decoding performance of the IT population is investigated (R1–R5 and scrambled stimuli only, see Materials and methods). *Figure 1d* also illustrates the SF classification accuracy across time in population-level representations. The peak accuracy is 24.68% higher than the chance level at 115 ms after the stimulus onset. These observations indicate the explicit presence of SF coding in the IT cortex. The strength of SF selectivity, considering the trial-to-trial variability, is provided in *Appendix 1—figure 1*, by ranking the SF bands for each neuron based on half of the

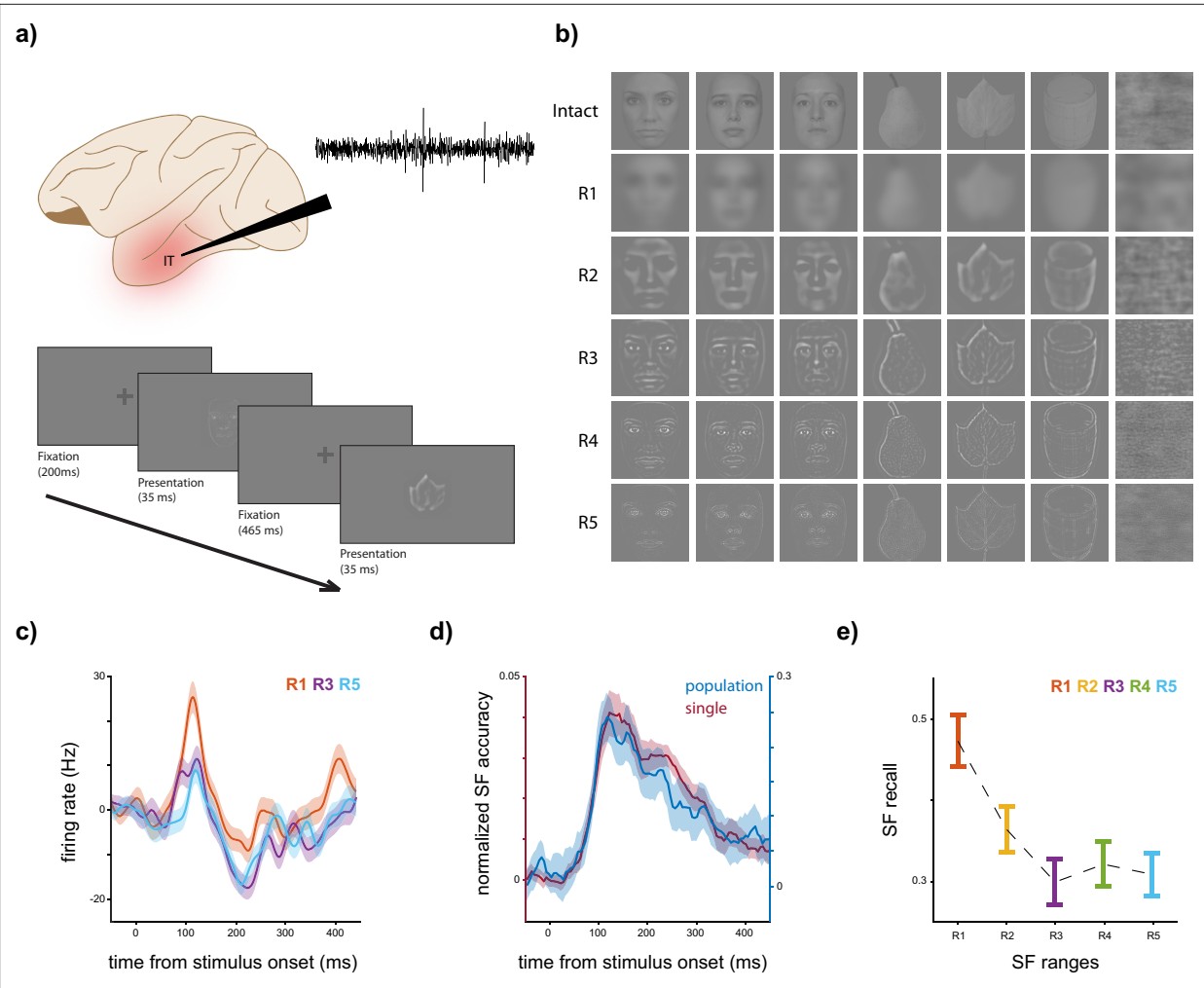

**Figure 1.** Experimental design and spatial frequency (SF) coding. (**a**) *Experimental design*. The design of the experiment involved the collection of responses from inferior temporal (IT) neurons to 15 stimuli (including six faces, three non-faces, and six selective stimuli, see Materials and methods) in six SF bands (intact and R1–R5, see Materials and methods), and two versions (scrambled and unscrambled) using a passive presentation task. The presentation of blocks starts if the monkey preserves fixation for 200 ms. Each block consisted of a 33-ms stimulus presentation followed by a blank screen with a fixation point of 465 ms, and each stimulus was presented 15 times. The recorded signals were sorted, and visually responsive neurons were selected (N = 266, see Materials and methods). (**b**) *A sample of the fixed stimulus set*. This panel shows three (out of six) faces, three non-faces, and one scrambled sample stimulus. Each row corresponds to an SF range starting with intact, followed by R1–R5 (low to high SF). (**c**) *A sample neuron*. The peristimulus time histogram (PSTH) of a sample neuron (N = 151, M1) for scrambled stimuli is depicted. To generate a response vector for a given stimulus or trial, the responses of each neuron were averaged in a 50-ms time window centered around the relevant time point. The PSTH was smoothed using a Gaussian kernel with a standard deviation of 20 ms. The responses of three SF bands (R1, R3, and R5) are shown for better illustration. (**d**) *SF coding exists in the IT cortex*. The decoding performance of SF ranges using scrambled stimuli is shown over time. Single- and population-level representations were fed into a linear discriminant analysis (LDA) algorithm to predict the SF range of the scrambled stimuli. Shadows illustrate the SEM and STD for single and population levels, respectively. This figure highlights the presence of SF coding in both individual and population neural activity. (**e**) *Low SF (LSF)-preferred nature of SF coding*. The population recall of each SF band in response to scrambled stimuli, determined using the LDA method, is presented. The error bars indicate the STD. The results demonstrate a decreasing trend as SF moves toward higher frequencies, suggesting a coarse-to-fine decoding preference.

The online version of this article includes the following source data for figure 1:

**Source data 1.** Source data for neuronal firing rates, SF decoding accuracy at single-unit and population levels, and recall of SF decoding per SF.

trials and then plotting the average responses for the obtained ranks for the other half of the trials. To determine the discrimination of each SF range, *Figure 1e* shows the recall of each SF content for the time window of 70–170 ms after stimulus onset. This observation reveals an LSF-preferred decoding behavior across the IT population (recall, R1 = 0.47 ± 0.04, R2 = 0.36 ± 0.03, R3 = 0.30 ± 0.03, R4 = 0.32 ± 0.04, R5 = 0.30 ± 0.03, and R1 > R5, p-value <0.001).

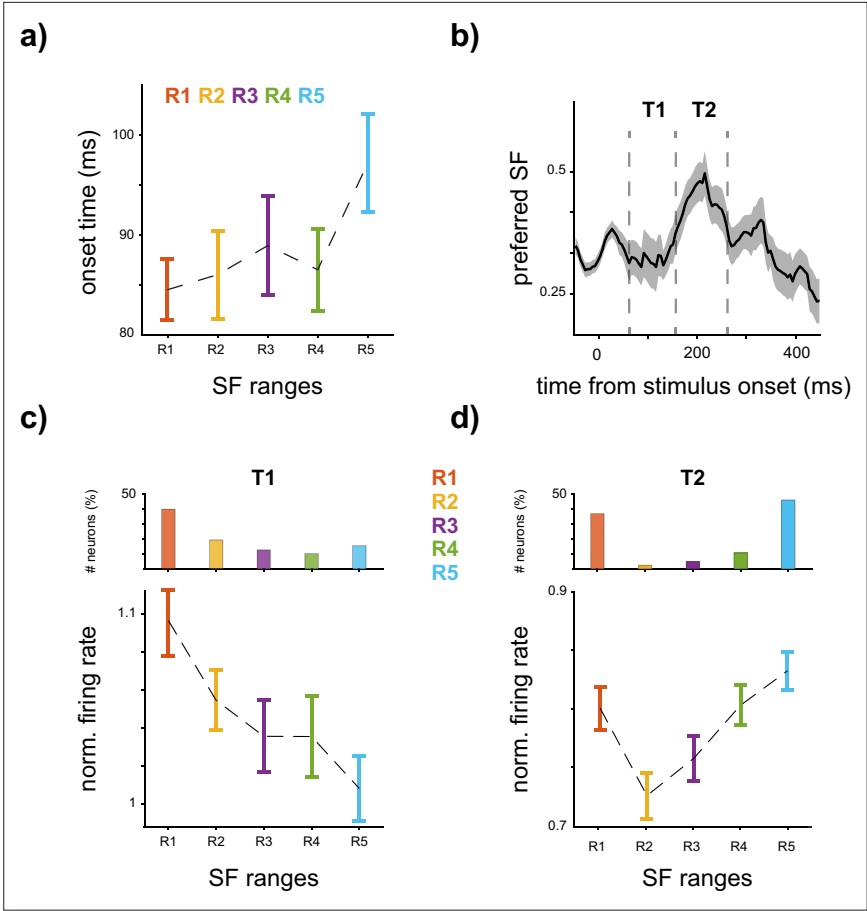

**Figure 2.** The temporal dynamics of spatial frequency (SF) representation. (**a**) *Course-to-fine nature of SF coding.* The onset time of the recall of each SF range in scrambled stimuli is illustrated, with error bars indicating the STD. The results suggest that the onset time of decoding increases as SF increases. (**b**) *SF preference shifts toward higher frequencies over time.* The time course of the average preferred SF (see Materials and methods) across neurons is illustrated. The average preferred SF of inferior temporal (IT) neurons moves toward higher frequencies from 170 ms after stimulus onset, reaching its highest value at 220 ms. A second peak emerges at 320 ms following the stimulus onset. The SF preference shows a monotonic increase followed by a decrease in time. The shadow shows SEM. (**c, d**) *Shift in neural response toward high SF (HSF).* The average response of all neurons within the two time intervals (T1 and T2 in panel b) is shown, with error bars indicating the SEM. (**c**) In T1, from 70 to 170 ms after stimulus onset, a decreasing response of the neurons is observed as the SF content shifts toward higher frequencies. The relative percentage of neurons showing stronger responses to SF ranges (R1–R5) in T1 is depicted in the inner top panel. R1 is the most responsive SF for roughly 40% of the neurons. (**d**) In the following interval (T2, 170–270 ms), an increasing tuning is observed from R2 to R5, where R5 elicits the highest firing rates. Furthermore, in T2, there is a roughly threefold increase in the percentage of neurons exhibiting stronger responses to R5 compared to T1, indicating a shift in the neurons' responses toward HSF (top panel).

The online version of this article includes the following source data for figure 2:

**Source data 1.** Source data for temporal dynamics of SF coding, including onset of recall, preferred SF, and neuronal responses across time intervals.

## Temporal dynamics of SF representation

The sample neuron and recall values in *Figure 1* indicate an LSF-preferred neuron response. To explore this behavior over time, we analyzed the temporal dynamics of SF representation. *Figure 2a* illustrates the onset of SF recalls, revealing a coarse-to-fine trend where R1 is decoded faster than R5 (onset times in milliseconds after stimulus onset, R1 = 84.5 ± 3.02, R2 = 86.0 ± 4.4, R3 = 88.9 ± 4.9, R4 = 86.5 ± 4.1, R5 = 97.15 ± 4.9, R1 < R5, p-value <0.001). *Figure 2b* illustrates the time course of the average preferred SF across the neurons. To calculate the preferred SF for each neuron, we multiplied the firing rate by the SF range and normalized the values (see Materials and methods). *Figure 2b* demonstrates

that following the early phase of the response (70–170 ms), the average preferred SF shifts toward HSF, reaching its peak at 215 ms after stimulus onset (preferred SF, 0.54 ± 0.15). Furthermore, a second peak emerges at 320 ms after stimulus onset (preferred SF, 0.22 ± 0.16), indicating a shift in the average preferred SF in the IT cortex toward higher frequencies. To analyze this shift, we divided the time course into two intervals of 70–170 ms, where the response peak of the neurons happens, and 170–270 ms, where the first peak of SF preference occurs. We calculated the percentage of the neurons that significantly responded to a specific SF range higher than others (one-way ANOVA with a significance level of 0.05, see Materials and methods) for the two time intervals. *Figure 2c, d* shows the percentage of the neurons in each SF range for the two time steps. In the early phase of the response (T1, 70–170 ms), the highest percentage of the neurons belongs to R1, 40.19%, and a decreasing trend is observed as we move toward higher frequencies (R1 = 40.19%, R2 = 19.60%, R3 = 13.72%, R4 = 10.78%, R5 = 15.68%). Moving to T2, the percentage of neurons responding to R1 higher than the others remains stable, dropping to 38.46%. The number of neurons in R2 also drops to under 5% from 19.60% observed in T1. On the other hand, the percentage of the neurons in R5 reaches 46.66% in T2 compared to 15.68% in T1 (higher than R1 in T1). This observation indicates that the increase in preferred SF is due to a substantial increase in the selective neurons to HSF, while the response of the neurons to R1 is roughly unchanged. To further understand the population response to various SF ranges, the average response across neurons for R1–R5 is depicted in *Figure 2c, d* (bottom panels). In the first interval, T1, an average LSF-preferred tuning is observed where the average neuron response decreases as the SF increases (normalized firing rate for R1 = 1.09 ± 0.01, R2 = 1.05 ± 0.01, R3 = 1.03 ± 0.01, R4 = 1.03 ± 0.02, R5 = 1.00 ± 0.01, Bonferroni corrected p-value for R2 < R5, 0.006). Considering the strength of responses to scrambled stimuli, the average firing rate in response to scrambled stimuli is 26.3 Hz, which is significantly higher than the response observed between −50 and 50 ms, where it is 23.4 Hz (p-value = $3 \times 10^{-5}$). In comparison, the mean response to intact face stimuli is 30.5 Hz, while non-face stimuli elicit an average response of 28.8 Hz. The distribution of neuron responses for scrambled, face, and non-face in T1 is illustrated in *Appendix 1—figure 2*. During the second time interval, excluding R1, the decreasing pattern transformed to an increasing one, with the response to R5 surpassing that of R1 (normalized firing rate for R1 = 0.80 ± 0.02, R2 = 0.73 ± 0.02, R3 = 0.76 ± 0.02, R4 = 0.81 ± 0.02, R5 = 0.84 ± 0.01, Bonferroni corrected p-value for R2 < R4, 0.022, R2 < R5, 0.0003, and R3 < R5, 0.03). Moreover, the average firing rates of scrambled, face, and non-face stimuli are 19.5, 19.4, and 22.4 Hz, respectively. The distribution of neuron responses is illustrated in *Appendix 1—figure 2*. These observations illustrate an LSF-preferred tuning in the early phase of the response, shifting toward HSF-preferred tuning in the late response phase.

## SF profile predicts category coding

Our findings indicate explicit SF coding in the IT cortex. Given the co-existence of SF and category coding in this region, we examine the relationship between SF and category codings. As depicted in *Figure 2*, while the average preferred SF across the neurons shifts to HSF, the most responsive SF range varies across individual neurons. To investigate the relation between SF representation and category coding, we identified an SF profile by fitting a quadratic curve to the neuron responses across SF ranges (R1–R5, phase-scrambled stimuli only). Then, according to the fitted curve, an SF profile is determined for each neuron (see Materials and methods). Five distinct profiles were identified based on the tuning curves (*Figure 3a*): (1) flat, where the neuron has no preferred SF (not included in the results), (2) LSF preferred (LP), where the neuron response decreases as SF increases, (3) HSF preferred (HP), where neuron response increases as the SF shifts toward higher SFs, (4) U-shaped where the neuron response to middle SF is lower than that of HSF or LSF, and (5) inverse U-shaped (IU), where the neuron response to middle SF is higher than that of LSF and HSF. The U-shaped and HSF-preferred profiles represent the largest and smallest populations, respectively. To check the robustness of the profiles, considering the trial-to-trial variability, the strength of SF selectivity in each profile is provided in *Appendix 1—figure 3*, by forming the profile of each neuron based on half of the trials and then plotting the average SF responses with the other half. Following profile identification, the object coding capability of each profile population is assessed. Here, instead of LDA, we employ the separability index (SI) introduced by *Dehaqani et al., 2016*, because of the LDA limitation in fully capturing the information differences between groups as it categorizes samples as correctly classified or misclassified.

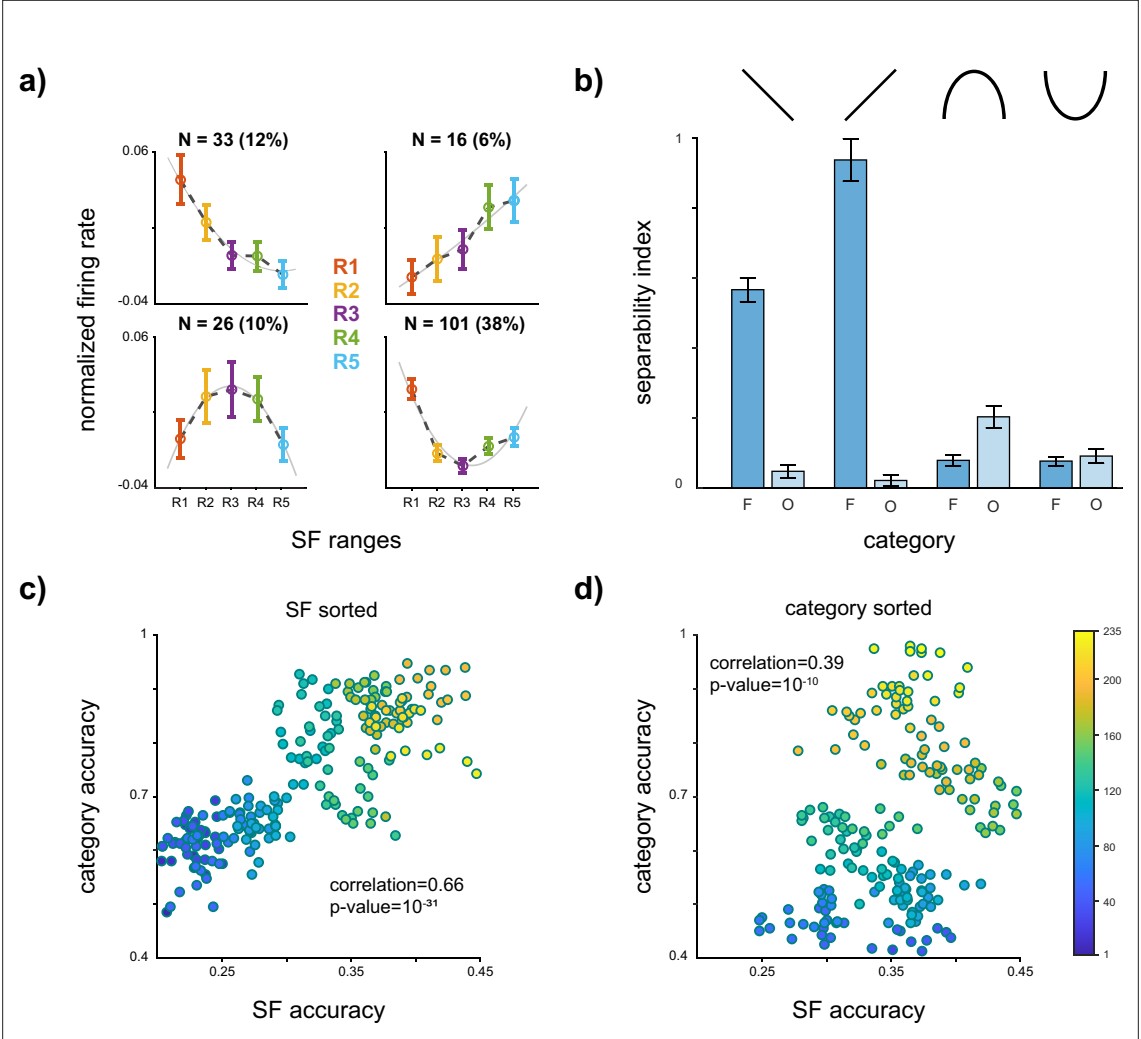

**Figure 3.** Spatial frequency (SF) profile predicts category coding. (**a, b**) *SF profile predicts category selectivity.* (**a**) The responses of each neuron were standardized by subtracting the mean and dividing by the standard deviation of the baseline time. Neurons were then categorized into four groups based on the fitting of a quadratic function to their responses (see Materials and methods). Each panel presents the average neuron responses within each category for SF ranges R1–R5, with error bars indicating the SEM of the response values. The percentage of the neurons in each category is displayed at the top of each panel. The 'flat' category, where the response to no SF was higher than others, was excluded from this analysis. (**b**) Separability index (SI) of face/non-face vs. scrambled stimuli is illustrated (see Materials and methods). The error bar shows STD. The SI value and SF profile are determined within the time window of 70–170 ms after stimulus onset. The high SF (HSF)-preferred population exhibited significantly higher face SI compared to the other groups. The low SF (LSF)-preferred population displayed a significant difference in face and non-face SI. On the other hand, the IU profile indicates a significantly higher SI value for the non-face compared to the face. The U-shaped profile did not show any significant differences between the face and the non-face. These results suggest that the neuron's response to various SF bands can predict its decoding capability. (**c, d**) *The relation between SF and category coding in sub-populations.* Initially, the linear discriminant analysis (LDA) method was employed to calculate the individual neuron's performance in the single-level category and SF coding. Next, a sorting procedure based on SF (panel c) and category (panel d) coding performances was conducted to create sub-populations of neurons exhibiting similar capabilities (see Materials and methods). The scatter plot of the category and SF coding accuracy of these sub-populations demonstrated a notably high degree of positive correlation between SF and category accuracies in the inferior temporal (IT) cortex.

The online version of this article includes the following source data for figure 3:

**Source data 1.** Source data for neuronal responses, SF profiles, and category selectivity, including separability indices and decoding accuracy for sub-populations of inferior temporal neurons.

To examine the face and non-face information separately, SI is calculated for face vs. scrambled and non-face vs. scrambled. *Figure 3a* displays the identified profiles, and *Figure 3b* indicates the average SI value during 70–170 ms after the stimulus onset. The HP profile shows significantly higher face information compared to other profiles (face SI for LP = 0.58 ± 0.03, HP = 0.89 ± 0.05, U = 0.07

± 0.01, IU = 0.07 ± 0.01, HP > LP, U, IU with p-value <0.001) and than non-face information in all other profiles (non-face SI for LP = 0.04 ± 0.01, HP = 0.02 ± 0.01, U = 0.19 ± 0.03, IU = 0.08 ± 0.02, and face SI in HP is greater than non-face SI in all profiles with p-value <0.001). This observation underscores the importance of middle and higher frequencies for face representation. The LSF-preferred profile also exhibits significantly higher face SI than non-face objects (p-value <0.001). On the other hand, in the IU profile, non-face information surpasses face SI (p-value <0.001), indicating the importance of middle frequency for the non-face objects. Finally, in the U profile, there is no significant difference between the face and non-face objects (face vs. non-face p-value = 0.36).

To assess whether the SF profiles distinguish category selectivity or merely evaluate the neuron's responsiveness, we quantified the number of face/non-face-selective neurons in the 70- to 170-ms time window. Our analysis shows a total of 43 face-selective neurons and 36 non-face-selective neurons (FDR-corrected p-value <0.05). The results indicate a higher proportion of face-selective neurons in LP and HP, while a greater number of non-face-selective neurons is observed in the IU category (number of face/non-face-selective neurons: LP = 13/3, HP = 6/2, IU = 3/9). The U category exhibits a roughly equal distribution of face and non-face-selective neurons (U = 14/13). This finding reinforces the connection between category selectivity and the identified profiles. We then analyzed the average neuron response to faces and non-faces within each profile. The difference between the firing rates for faces and non-faces in none of the profiles is significant (face/non-face average firing rate (Hz): LP = 36.72/28.77, HP = 28.55/25.52, IU = 21.55/27.25, U = 38.48/36.28, rank-sum with significance level of 0.05). Although the observed differences are not statistically significant, they provide support for the association between profiles and categories rather than mere responsiveness.

Next, to examine the relation between the SF (category) coding capacity of the single neurons and the category (SF) coding capability of the population level, we calculated the correlation between coding performance at the population level and the coding performance of single neurons within that population (*Figure 3a* and *Figure 3b*). In other words, we investigated the relation between single and population levels of coding capabilities between SF and category. The SF (or category) coding performance of a sub-population of 20 neurons that have roughly the same single-level coding capability of the category (or SF) is examined. Neurons were sorted based on their SF or category performances, resulting in two separate groups of ranks—one for SF and another for category. Subsequently, we selected sub-populations of neurons with similar ranks according to SF or category (see Materials and methods). Each sub-population comprises 20 neurons with approximately similar SF (or category) performance levels. Then, the SF and category decoding accuracy is calculated for each sub-population. The scatterplot of individual vs. sub-population accuracy demonstrated a significant positive correlation between the sub-population performance and the accuracy of individual neurons within those populations. Specifically, the correlation value for SF- and category-sorted groups is 0.66 (p-value = $10^{-31}$) and 0.39 (p-value = $10^{-10}$), respectively. This observation illustrates that SF coding capacity at single-level representations significantly predicts category coding capacity at the population level.

## Uncorrelated mechanisms for SF and category coding

As both SF and category coding exist in the IT cortex at both the single-neuron and population levels, we investigated their underlying coding mechanisms (for single and population levels separately). *Figure 4a* displays the scatter plot of SF and category coding capacity for individual neurons. The correlation between SF and category accuracy across individual neurons shows no significant relationship (correlation: 0.024 and p-value: 0.53), suggesting two uncorrelated mechanisms for SF and category coding. To explore the population-level coding, we considered neuron weights in the LDA classifier as indicators of each neuron's contribution to population coding. *Figure 4b* indicates the scatter plot of the neuron's weights in SF and category decoding. The LDA weights reveal no correlation between the patterns of neuron contribution in population decoding of SF and category (correlation = 0.002 and p-value = 0.39). These observations indicate uncorrelated coding mechanisms for SF and category in both single and population-level representations in the IT cortex. *Figures 3 and 4* examine different aspects of the relationship between SF and category coding. *Figure 3* highlights a relationship between SF coding at the single-neuron level and category coding at the population level. Conversely, *Figure 4* demonstrates the uncorrelated mechanisms underlying SF and category coding, showing that a neuron's ability to decode SF is not predictive of its ability to decode object

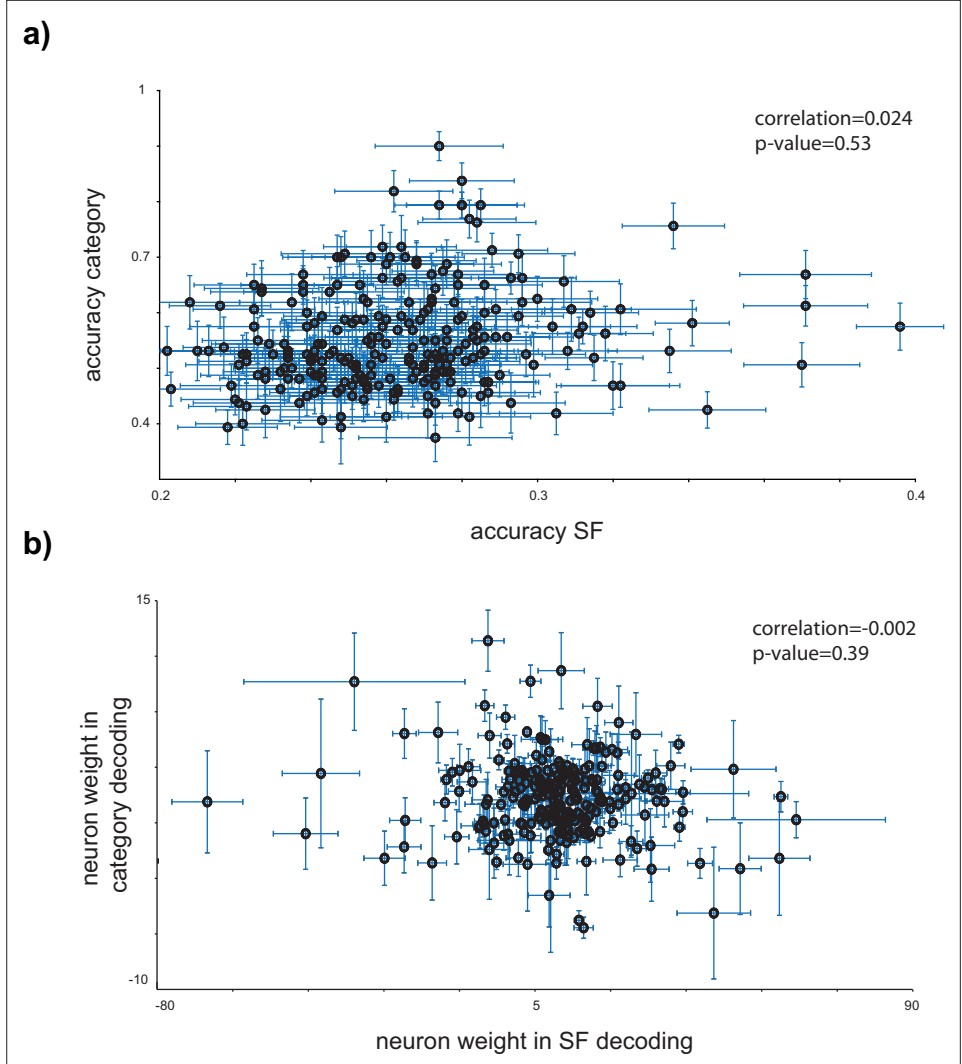

**Figure 4.** Uncorrelated mechanisms for spatial frequency (SF) and category coding. (**a**) *Uncorrelated SF and category coding in the single level.* The scatter plot indicates the category–SF accuracies and does not reveal a significant correlation between SF and category coding capabilities within the inferior temporal (IT) cortex at the single-neuron level. The error bars show the STD for SF and category decoding accuracies. (**b**) *Uncorrelated neuron contribution in SF and category coding in population.* The linear discriminant analysis (LDA) weight of each neuron is considered as the neuron contribution in the population coding of SF or category (see Materials and methods). The scatter plot of the neuron weights in SF shows a near-zero correlation with the neuron weights in category coding.

The online version of this article includes the following source data for figure 4:

**Source data 1.** Source data for single-neuron and population-level contributions to SF and category coding.

categories. This distinction underscores that while SF and category coding are related at broader levels, their underlying mechanisms are independent, emphasizing the distinct processes driving each form of coding.

Next, to investigate SF and category coding characteristics, we systematically removed individual neurons from the population and measured the resulting drop in LDA classifier accuracy as a metric for the neuron's impact, termed single-neuron contribution (SNC). *Figure 5a* illustrates the SNC score for SF (two labels, LSF (R1 and R2) vs. HSF (R4 and R5)) and category (face vs. non-face) decoding within 70–170 ms after the stimulus onset. The SNC in SF is significantly higher than for category (average SNC for SF = 0.51% ± 0.02 and category = 0.1% ± 0.04, SF > category with p-value = 1.6 × $1.6^{-13}$). Therefore, SF representation relies more on individual neuron representations, suggesting a

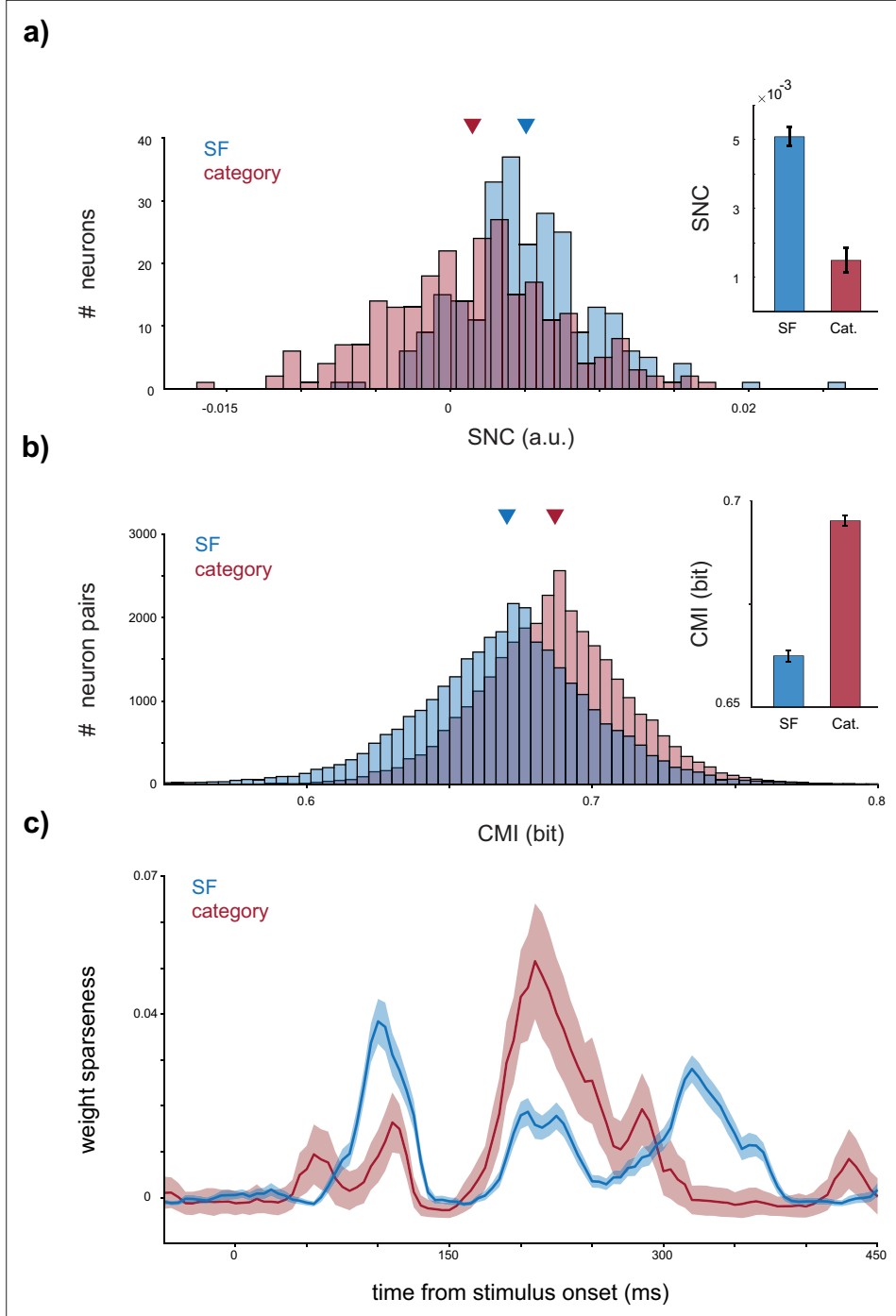

**Figure 5.** Sparse spatial frequency (SF) coding compared to category coding. (**a, b**) *Sparse mechanism for SF coding.* (**a**) The contribution of each neuron in SF and category (face vs. non-face) decoding is evaluated by removing it from the feature set fed to the linear discriminant analysis (LDA) within the time window of 70–170 ms after stimulus onset. The histogram of the single-neuron contribution (SNC) value (see Materials and methods) is presented, indicating the amount of accuracy loss when a neuron is removed. The bar plot displays the average SNC values for SF and category, with error bars representing the SEM. The SNC value for SF is significantly higher than for the category. (**b**) Furthermore, the conditional mutual information (CMI) of each neuron pair, conditioned to the label (category or SF), is illustrated. CMI reflects the information redundancy between neuron pairs during SF or category decoding. A lower CMI value for SF indicates that individual neurons carry more independent SF-related information compared to category information. (**c**) *Sparse neuron contribution in SF coding at the early phase of the response.* To investigate the contribution of the neurons in population decoding, the sparseness of

*Figure 5 continued on next page*

*Figure 5 continued*

the LDA weights assigned to each neuron is calculated. Higher sparseness indicates a greater contribution of a smaller group of neurons to the decoding process. The time course of weight sparseness is depicted for SF and category (face vs. non-face) decoding, with shadows representing the STD. During the early phase of the response, the sparseness of SF-related weights is higher than that of the category, while this relationship is reversed during the late phase of the response.

The online version of this article includes the following source data for figure 5:

**Source data 1.** Source data for single-neuron and population contributions to SF and category decoding, including single-neuron contribution values, conditional mutual information, and temporal sparseness of LDA weights.

---

sparse mechanism of SF coding where single-level neuron information is less redundant. In contrast, single-level representations of category appear to be more redundant and robust against information loss or noise at the level of individual neurons. We utilized conditional mutual information (CMI) between pairs of neurons conditioned on the label, SF (LSF (R1 and R2) vs. HSF (R4 and R5)) or category, to assess the information redundancy across the neurons. CMI quantifies the shared information between the population of two neurons regarding SF or category coding. *Figure 5b* indicates a significantly lower CMI for SF (average CMI for SF = 0.66 ± 0.0009 and category = 0.69 ± 0.0007, SF < category with p-value ≈ 0), indicating that neurons carry more independent SF-related information than category-related information.

To investigate each neuron's contribution to the decoding procedure (LDA decision), we computed the sparseness of the LDA weights corresponding to each neuron (see Materials and methods). For SF, we trained the LDA on R1, R2, R4, and R5 with two labels (one for R1 and R2 and the alternative for R4 and R5). A second LDA was trained to discriminate between faces and non-faces. Subsequently, we calculated the sparseness of the weights associated with each neuron in SF and category decoding. *Figure 5c* illustrates the time course of the weight sparseness for SF and category. The category reflects a bimodal curve with the first peak at 110 ms and the second at 210 ms after stimulus onset. The second peak is significantly larger than the first one (category first peak, 0.016 ± 0.007, second peak, 0.051 ± 0.013, and p-value <0.001). In SF decoding, neurons' weights exhibit a trimodal curve with peaks at 100, 215, and 320 ms after the stimulus onset. The first peak is significantly higher than the other two (SF first peak, 0.038 ± 0.005, second peak, 0.018 ± 0.003, third peak, 0.028 ± 0.003, first peak > second peak with p-value <0.001, and first peak > third peak with p-value = 0.014). Comparing SF and category, during the early phase of the response (70–170 ms), SF sparseness is higher, while in 170–270 ms, the sparseness value of the category is higher (p-value <0.001 for both time intervals). This suggests that, initially, most neurons contribute to category representation, but later, the majority of neurons are involved in SF coding. These findings support distinct mechanisms governing SF and category coding in the IT cortex.

## SF representation in the artificial neural networks

We conducted a thorough analysis to compare our findings with CNNs. To assess the SF coding capabilities and trends of CNNs, we utilized popular architectures, including ResNet18, ResNet34, VGG11, VGG16, InceptionV3, EfficientNetb0, CORNet-S, CORTNet-RT, and CORNet-z, with both pre-trained on ImageNet and randomly initialized weights (see Materials and methods). Employing feature maps from the four last layers of each CNN, we trained an LDA model to classify the SF content of input images. *Figure 6a* shows the SF decoding accuracy of the CNNs on our dataset (SF decoding accuracy with random (R) and pre-trained (P) weights, ResNet18: P = 0.96 ± 0.01/R = 0.94 ± 0.01, ResNet34: P = 0.95 ± 0.01/R = 0.86 ± 0.01, VGG11: P = 0.94 ± 0.01/R = 0.93 ± 0.01, VGG16: P = 0.92 ± 0.02/R = 0.90 ± 0.02, InceptionV3: P = 0.89 ± 0.01/R = 0.67 ± 0.03, EfficientNetb0: P = 0.94 ± 0.01/R = 0.30 ± 0.01, CORNet-S: P = 0.77 ± 0.02/R = 0.36 ± 0.02, CORTNet-RT: P = 0.31 ± 0.02/R = 0.33 ± 0.02, and CORNet-z: P = 0.94±0.01/R = 0.97 ± 0.01). Except for CORNet-z, object recognition training increases the network's capacity for SF coding, with an improvement as significant as 64% in EfficientNetb0. Furthermore, except for the CORNet family, LSF content exhibits higher recall values than HSF content, as observed in the IT cortex (p-value with random (R) and pre-trained (P) weights, ResNet18: P = 0.39/R = 0.06, ResNet34: P = 0.01/R = 0.01, VGG11: P = 0.13/R = 0.07, VGG16: P = 0.03/R = 0.05, InceptionV3: P = <0.001/R = 0.05, EfficientNetb0: P = 0.07/R = 0.01). The recall

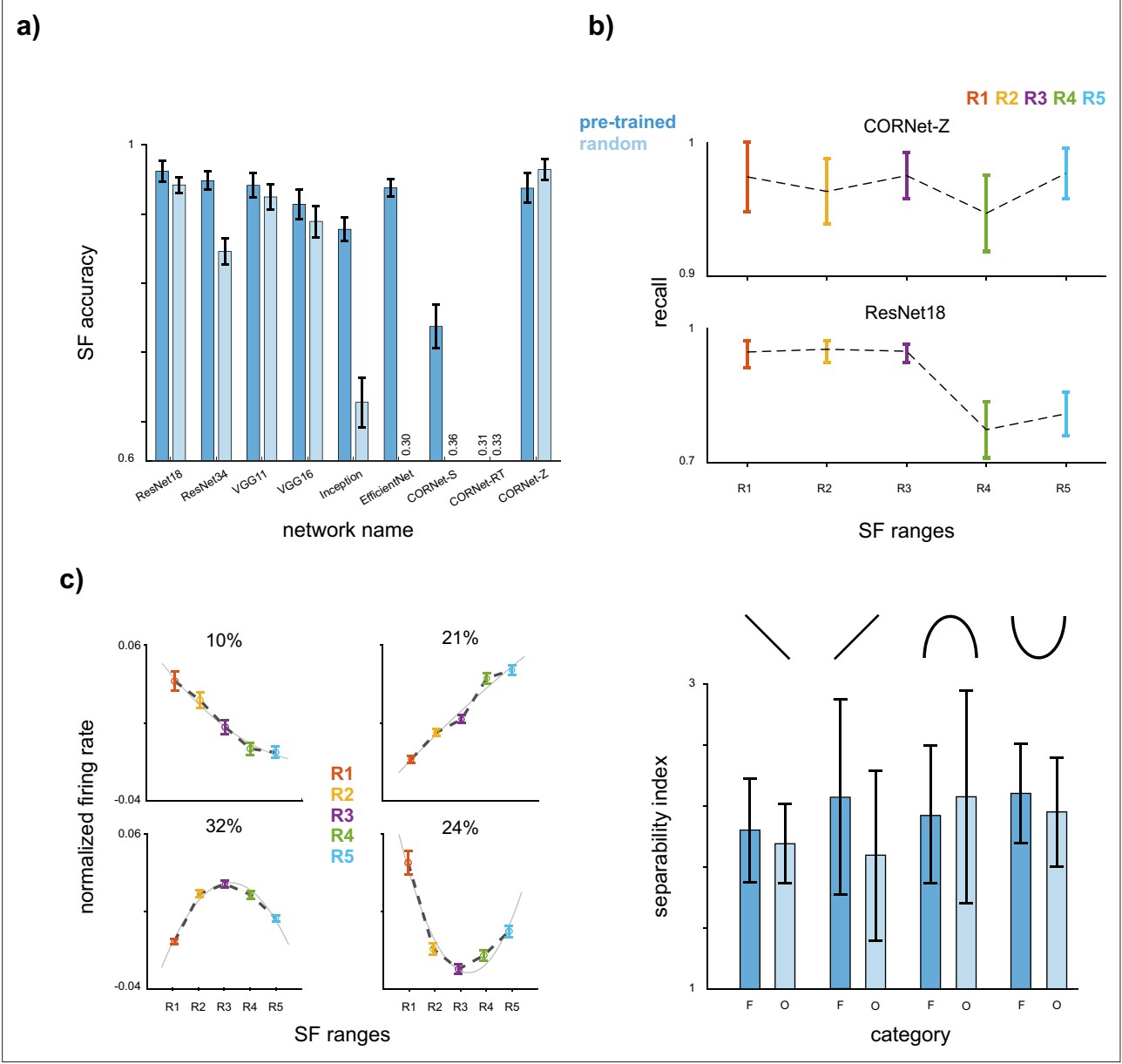

**Figure 6.** Spatial frequency (SF) representation in convolutional neural networks (CNNs). (**a**) *SF coding capabilities*. We assessed the SF coding capabilities of popular CNN architectures (ResNet18, ResNet34, VGG11, VGG16, InceptionV3, EfficientNetb0, CORNet-S, CORTNet-RT, and CORNet-z) using both randomly initialized (R) and pre-trained (P) weights on ImageNet. A linear discriminant analysis (LDA) model was trained using feature maps from the four last layers of each CNN to classify the SF content of input images. The SF decoding accuracy for each CNN on our dataset is presented with error bars indicating the STD. (**b**) *Low SF (LSF)-preferred recall performance*. The recall performance of two sample networks (CORNET-z and ResNet18) is presented. STD values are illustrated with error bars. The recall values for LSF content were higher than high SF (HSF) content in most CNNs, resembling the trends observed in the inferior temporal (IT) cortex. (**c**) The profiles (left) and face/non-face separability index (SI) value (right) of a sample network (ResNet18). Error bars show STD. Profiles are calculated similarly to the IT cortex. CNNs did not replicate the SF-based profiles observed in the IT cortex.

The online version of this article includes the following source data for figure 6:

**Source data 1.** Source data for SF decoding and recall performance in multiple CNN architectures, SF preference profiles and face/non-face separability indices.

values of CORNet-Z and ResNet18 are illustrated in *Figure 6b*. However, while the CNNs exhibited some similarities in SF representation with the IT cortex, they did not replicate the SF-based profiles that predict neuron category selectivity. As depicted in *Figure 6c*, although neurons formed similar profiles, these profiles were not associated with the category decoding performances of the neurons sharing the same profile.

## Discussion

Utilizing neural responses from the IT cortex of passive-viewing monkeys, we conducted a study on SF representation within this pure visual high-level area. Numerous psychophysical studies have underscored the significant impact of SF on object recognition, highlighting the importance of its representation. To the best of our knowledge, this study presents the first attempt to systematically examine the SF representation in a high-level area, that is, the IT cortex, using extracellular recording. Understanding SF representation is crucial, as it can elucidate the object recognition procedure in the IT cortex.

Our findings demonstrate explicit SF coding at both the single-neuron and population levels, with LSF being decoded faster and more accurately than HSF. During the early phase of the response, we observe a preference for LSF, which shifts toward a preference for HSF during the late phase. Next, we made profiles based on SF-only (phase-scrambled stimuli) responses for each neuron to predict its category selectivity. Our results show a direct relationship between the population's category coding capability and the SF coding capability of individual neurons. While we observed a relation between SF and category coding, we have found uncorrelated representations. Unlike category coding, SF relies more on sparse, individual neuron representations. Finally, when comparing the responses of IT with those of CNNs, it is evident that while SF coding exists in CNNs, the SF profile observed in the IT cortex is notably absent. Our results are based on grouping the neurons of the two monkeys; however, the results remain consistent when looking at the data from individual monkeys as illustrated in Appendix 2. However, for neurons preferring LSF, we observed some inconsistencies across monkeys, which may reflect individual differences or sampling variability. These findings highlight the complexity of SF processing in the IT cortex and suggest the need for further research to explore these variations.

The influence of SF on object recognition has been extensively investigated through psychophysical studies (*Ashtiani et al., 2017*; *Caplette et al., 2014*; *Cheung and Bar, 2014*; *Craddock et al., 2013*; *Joubert et al., 2007*; *Schyns and Oliva, 1994*). One frequently explored theory is the coarse-to-fine nature of SF processing in object recognition (*Gao and Bentin, 2011*; *Kauffmann et al., 2015*; *Rokszin et al., 2016*; *Rotshtein et al., 2010*; *Schyns and Oliva, 1994*; *Yardley et al., 2012*). This aligns with our observation that the onset of LSF is significantly lower than HSF. Different SF bands carry distinct information, progressively conveying coarse-to-fine shape details as we transition from LSF to HSF. Psychophysical studies have indicated the utilization of various SF bands for distinct categorization tasks (*Rotshtein et al., 2010*). Considering the face as a behaviorally demanded object, psychophysical studies have observed the influence of various SF bands on face recognition. These studies consistently show that enhanced face recognition performance is achieved in the middle and higher SF bands compared to LSF (*Awasthi et al., 2012*; *Cheung et al., 2008*; *Costen et al., 1996*; *Fiorentini et al., 1983*; *Hayes et al., 1986*; *Jeantet et al., 2019*). These observations resonate with the identified SF profiles in our study. Neurons that exhibit heightened responses as SF shifts toward HSF demonstrate superior coding of faces compared to other neuronal groups.

Unlike psychophysical studies, imaging studies in this area have been relatively limited. *Gaska et al., 1988* observed low-pass tuning curves in the V3A area, and *Chen et al., 2018* reported an average low-pass tuning curve in the superior colliculus (SC). *Purushothaman et al., 2014* identified two distinct types of neurons in V1 based on their response to SF. The majority of neurons in the first group exhibited a monotonically shifting preference toward HSF over time. In contrast, the second group showed an initial increase in preferred SF followed by a decrease. Our findings align with these observations, showing a rise in preferred SF starting at 170 ms after stimulus onset, followed by a decline at 220 ms after stimulus onset. Additionally, *Zhang et al., 2023* found that LSF is the preferred band for over 40% of V4 neurons. This finding is also consistent with our observations, where approximately 40% of neurons consistently exhibited the highest firing rates in response to LSF throughout all response phases. Collectively, these results suggest that the average LP tuning curve observed in

the IT cortex could be a characteristic inherited from the lower areas in the visual hierarchy. Moreover, examining the course-to-fine theory of SF processing, *Chen et al., 2018* and *Purushothaman et al., 2014* observed a faster response to LSF compared to HSF in SC and V1, which resonates with our course-to-fine observation in SF decoding. When analyzing the relationship between the SF content of complex stimuli and IT responses, *Bermudez et al., 2009* observed a correlation between neural responses in the IT cortex and the SF content of the stimuli. This finding is in line with our observations, as decoding results directly from the distinct patterns exhibited by various SF bands in neural responses.

To rule out the degraded contrast sensitivity of the visual system to medium and high SF information because of the brief exposure time, we repeated the analysis with 200 ms exposure time as illustrated in *Appendix 1—figure 4* which indicates the same LSF-preferred results. Furthermore, according to *Figure 2*, the average firing rate of IT neurons for HSF could be higher than LSF in the late response phase. It indicates that the amount of HSF input received by the IT neurons in the later phase is as much as LSF; however, its impact on the IT response is observable in the later phase of the response. Thus, the LSF preference is because of the temporal advantage of the LSF processing rather than contrast sensitivity. Next, according to *Figure 3a*, 6% of the neurons are HSF-preferred and their firing rate in HSF is comparable to the LSF firing rate in the LSF-preferred group. This analysis is carried out in the early phase of the response (70–170 ms). While most of the neurons prefer LSF, this observation shows that there is an HSF input that excites a small group of neurons. Importantly, findings involving small neuronal populations can still be meaningful, as studies like *Dalgleish et al., 2020* have demonstrated that perception can arise from the activity of as few as 14 neurons in the mouse cortex, emphasizing the robustness of sparse coding. Additionally, the highest SI belongs to the HSF-preferred profile in the early phase of the response, which supports the impact of the HSF part of the input. Similar LSF-preferred responses are also reported by *Chen et al., 2018* (50 ms for SC) and (*Zhang et al., 2023*) (3.5–4 s for V2 and V4). Therefore, our results show that the LSF-preferred nature of the IT responses in terms of firing rate and recall is not due to the weakness or lack of input source (or information) for HSF but rather to the processing nature of the SF in the IT cortex.

*Hong et al., 2016* suggested that the neural mechanisms responsible for developing tolerance to identity-preserving transform also contribute to explicitly representing these category–orthogonal transforms, such as rotation. Extending this perspective to SF, our results similarly suggest an explicit representation of SF within the IT population. However, unlike transforms such as rotation, the neural mechanisms in IT leverage various SF bands for various categorization tasks. Furthermore, our analysis introduced a novel SF-only profile for the first time predicting category selectivity.

These findings prompt the question of why the IT cortex explicitly represents and codes the SF content of the input stimuli. In our perspective, the explicit representation and coding of SF contents in the IT cortex facilitates object recognition. The population of the neurons in the IT cortex becomes selective for complex object features, combining SFs to transform simple visual features into more complex object representations. However, the specific mechanism underlying this combination is yet to be known. The diverse SF contents present in each image carry valuable information that may contribute to generating expectations in predictive coding during the early phase, thereby facilitating information processing in subsequent phases. This top–down mechanism is suggested by the works of *Bar, 2003* and *Fenske et al., 2006*.

Moreover, each object has a unique 'characteristic SF signature', representing its specific arrangement of SFs. 'Characteristic SF signatures' refer to the unique patterns or profiles of SFs associated with different objects or categories of objects. When we look at visual stimuli, such as objects or scenes, they contain specific arrangements of different SFs. Imagine a scenario where we have two objects, such as a cat and a car. These objects will have different textures and shapes, which correspond to different distributions of SFs. The cat, for instance, might have a higher concentration of mid-range SFs related to its fur texture, while the car might have more pronounced LSFs that represent its overall shape and structure. The IT cortex encodes these signatures, facilitating robust discrimination and recognition of objects based on their distinctive SF patterns.

The concept of 'characteristic SF signatures' is also related to the 'SF tuning' observed in our results. Neurons in the visual cortex, including the IT cortex, have specific tuning preferences for different SFs. Some neurons are more sensitive to HSF, while others respond better to LSF. This distribution of sensitivity allows the visual system to analyze and interpret different information related to different

SF components of visual stimuli concurrently. Moreover, the IT cortex's coding of SF can contribute to object invariance and generalization. By representing objects in terms of their SF content, the IT cortex becomes less sensitive to variations in size, position, or orientation, ensuring consistent recognition across different conditions. SF information also aids the IT cortex in categorizing objects into meaningful groups at various levels of abstraction. Neurons can selectively respond to shared SF characteristics among different object categories (assuming that objects in the same category share a level of SF characteristics), facilitating decision-making about visual stimuli. Overall, we posit that SF's explicit representation and coding in the IT cortex enhance its proficiency in object recognition. By capturing essential details and characteristics of objects, the IT cortex creates a rich representation of the visual world, enabling us to perceive, recognize, and interact with objects in our environment.

Finally, we compared SF's representation trends and findings within the IT cortex and the current state of the art networks in deep neural networks. CNNs stand as one of the most promising models for comprehending visual processing within the primate ventral visual processing stream (*Kubilius, 2019*; *Kubilius et al., 2018*). Examining the higher layers of CNN models (most similar to IT), we found that randomly initialized and pre-trained CNNs can code for SF. This is consistent with our previous work on the CIFAR dataset (*Toosi et al., 2022*). Nevertheless, they do not exhibit the SF profile we observed in the IT cortex. This emphasizes the uniqueness of SF coding in the IT cortex and suggests that artificial neural networks might not fully capture the complete complexity of biological visual processing mechanisms, even when they encompass certain aspects of SF representation. Our results intimate that the IT cortex uses a different mechanism for SF coding compared to contemporary deep neural networks, highlighting the potential for innovating new approaches to consider the role of SF in the ventral stream models.

Our results are not affected by several potential confounding factors. First, each stimulus in the set also has a corresponding phase-scrambled variant. These phase-scrambled stimuli maintain the same SF characteristics as their respective face or non-face counterparts but lack shape information. This approach allows us to investigate SF representation in the IT cortex without the confounding influence of shape information. Second, our results, obtained through a passive-viewing task, remain unaffected by attention mechanisms. Third, all stimuli (intact, SF filtered, and phase scrambled) are corrected for illumination and contrast to remove the attribution of the category–orthogonal basic characteristics of stimuli into the results (see Materials and methods). Fourth, while our dataset does not exhibit a balance in samples per category, it is imperative to acknowledge that this imbalance does not exert an impact on our observed outcomes. We have equalized the number of samples per category when training our classification models by random sampling from the stimulus set (see Materials and methods). One limitation of our study is the relatively low number of objects in the stimulus set. However, the decoding performance of category classification (face vs. non-face) in intact stimuli is 94.2%. The recall value for objects vs. scrambled is 90.45%, and for faces vs. scrambled is 92.45 (p-value = 0.44), which indicates the high level of generalizability and validity characterizing our results. Finally, since our experiment maintains a fixed SF content in terms of both cycles per degree and cycles per image, further experiments are needed to discern whether our observations reflect sensitivity to cycles per degree or cycles per image.

In summary, we studied the SF representation within the IT cortex. Our findings reveal the existence of a sparse mechanism responsible for encoding SF in the IT cortex. Moreover, we studied the relationship between SF representation and object recognition by identifying an SF profile that predicts object recognition performance. These findings establish neural correlates of the psychophysical studies on the role of SF in object recognition and shed light on how IT represents and utilizes SF for the purpose of object recognition.

# Materials and methods

**Key resources table**

| Reagent type (species) or resource | Designation | Source or reference | Identifiers | Additional information |
|---|---|---|---|---|
| Biological sample (*Macaca mulatta*, male) | IT cortex neurons; Neurons; Monkey | This paper | | From two adult male macaques (10 and 11 kg); see Materials and methods |
| Software, algorithm | MATLAB | MathWorks | | Used for stimulus presentation, control, and analysis |
| Software, algorithm | MonkeyLogic toolbox | MonkeyLogic website | | For experimental control in MATLAB |
| Software, algorithm | Python | Python Software Foundation | 3.10 | Used for data analysis and machine learning workflows |
| Software, algorithm | PyTorch | pytorch.org | 2.0 | Deep learning framework used for neural modeling |

## Animals and recording

The activity of neurons in the IT cortex of two male macaque monkeys weighing 10 and 11 kg, respectively, was analyzed following the National Institutes of Health Guide for the Care and Use of Laboratory Animals and the Society for Neuroscience Guidelines and Policies. The experimental procedures were approved by the Institute of Fundamental Science committee (Approval Code: 99/60/1/160/1). Before implanting a recording chamber in a subsequent surgery, magnetic resonance imaging and computed tomography scans were performed to locate the prelunate gyrus and arcuate sulcus. The surgical procedures were carried out under sterile conditions and isoflurane anesthesia. Each monkey was fitted with a custom-made stainless-steel chamber, secured to the skull using titanium screws and dental acrylics. A craniotomy was performed within the 30 × 70 mm chamber for both monkeys, with dimensions ranging from 5 to 30 mm A/P and 0 to 23 mm M/L.

During the experiment, the monkeys were seated in custom-made primate chairs, and their heads were restrained while a tube delivered juice rewards to their mouths. The system was mounted in front of the monkey, and eye movements were captured at 2 kHz using the EyeLink PM-910 Illuminator Module and EyeLink 1000 Plus Camera (SR Research Ltd, Ottawa, CA). Stimulus presentation and juice delivery were controlled using custom software written in MATLAB with the MonkeyLogic toolbox (see Supplementary File – *PreTestCode.m*). Visual stimuli were presented on a 24-inch LED-lit monitor (AsusVG248QE, 1920 × 1080, 144 Hz) positioned 65.5 cm away from the monkeys' eyes. The actual time the stimulus appeared on the monitor was recorded using a photodiode (OSRAM Opto Semiconductors, Sunnyvale, CA).

One electrode was affixed to a recording chamber and positioned within the craniotomy area using the Narishige two-axis platform, allowing for continuous electrode positioning adjustment. To make contact with or slightly penetrate the dura, a 28-gauge guide tube was inserted using a manual oil hydraulic micromanipulator from Narishige, Tokyo, Japan. For recording neural activity extracellularly in both monkeys, varnish-coated tungsten microelectrodes (FHC, Bowdoinham, ME) with a shank diameter of 200–250 μm and impedance of 0.2–1 M$\omega$ (measured at 1 kHz) were inserted into the brain. A pre-amplifier and amplifier (Resana, Tehran, Iran) were employed for single-electrode recordings, with filtering set between 300 Hz and 5 kHz for spikes and 0.1 Hz and 9 kHz for local field potentials. Spike waveforms and continuous data were digitized and stored at a sampling rate of 40 kHz for offline spike sorting and subsequent data analysis. Area IT was identified based on its stereotaxic location, position relative to nearby sulci, patterns of gray and white matter, and response properties of encountered units.

## Stimulus and task paradigm

The experimental task comprised two distinct phases: selectivity and main phases, each involving different stimuli. During the selectivity phase, the objective was to identify a responsive neuron for recording purposes. If an appropriate neuron was detected, the main phase was initiated. However, if a responsive neuron was not observed, the recording location was adjusted, and the selectivity phase

was repeated. First, we will outline the procedure for stimulus construction, followed by an explanation of the task paradigm.

## The stimulus set

The size of each image was 500 × 500 pixels. Images were displayed on a 120-Hz monitor with a resolution of 1920 × 1080 pixels. The monitor's response time (changing the color of pixels in gray space) was one millisecond. The monkey's eyes were located at a distance of 65 cm from the monitor. Each stimulus occupied a space of 5 × 5 degrees. All images were displayed in the center of the monitor. During the selectivity phase, a total number of 155 images were used as stimuli. Regarding SF, the stimuli were divided into unfiltered and filtered categories. Unfiltered images included 74 separate grayscale images in the categories of animal face, animal body, human face, human body, man-made, and natural. To create the stimulus, these images were placed on a gray background with a value of 0.5. The filtered images included 27 images in the same categories as the previous images, which were filtered in two frequency ranges (along with the intact form): low (1–10 cycles per image) and high (18–75 cycles per image), totaling 81 images. In the main phase of the test, nine images, including three non-face images and six face images, were considered. These images were displayed in the center of the monitor. During the selectivity phase, a total number of 155 images were used as stimuli. Regarding SF, the stimuli were divided into unfiltered and filtered categories. Unfiltered images included 74 separate grayscale images in the categories of animal face, animal body, human face, human body, man-made, and natural. To create the stimulus, these images were placed on a gray background with a value of 0.5. The filtered images included 27 images in the same categories as the previous images, which were filtered in two frequency ranges (along with the intact form): low (1–10 cycles per image) and high (18–75 cycles per image), totaling 81 images. In the main phase of the test, nine images, including three non-face images and six face images, were considered. These images were displayed in *Figure 1c*. For the main phase, the images were filtered in five frequency ranges. These intervals were 1–5, 5–10, 10–18, 18–45, and 45–75 cycles per image. For each image in each frequency range, a scrambled version had been obtained by scrambling the image phase in the Fourier transform domain. Therefore, each image in the main phase contained one unfiltered version (intact), five filtered versions (R1–R5), and six scrambled versions (i.e., 12 versions in total).

## SF filtering

Butterworth filters were used to filter the images in this study. A low-pass Butterworth filter is defined as follows:

$$B_{lp}(r,f_c) = \frac{1}{1 + (r/f_c)^{2n}} \tag{1}$$

where $B_{lp}$ is the absolute value of the filter, $r$ is the distance of the pixel from the center of the image, $f_c$ is the filter's cutoff frequency in terms of cycles per image, and $n$ is the order of the filter. Similarly, the high-pass filter is defined as follows:

$$B_{hp}(r,f_c) = \frac{1}{1 + (f_c/r)^{2n}} \tag{2}$$

To create a band-pass filter with a pass frequency of $f_1$ and a cutoff frequency of $f_2$, a multiplication of a high-pass and a low-pass filter was performed ($B_{bp}(r,f_1,f_2) = B_{lp}(r;f_1) \times B_{hp}(r;f_2)$). To apply the filter, the image underwent a two-dimensional Fourier transform, followed by multiplication with the appropriate filter. Subsequently, the inverse Fourier transform was employed to obtain the filtered image. Afterward, a linear transformation was applied to adjust the brightness and contrast of the images. Brightness was determined by the average pixel value of the image, while contrast was represented by its standard deviation (STD). To achieve specific brightness (L) and contrast (C) levels, the following equation was employed to correct the images:

$$I_{norm} = C \times \left( \frac{I - \mu}{\sigma} \right) + L \tag{3}$$

where $\sigma$ and $\mu$ are the STD and mean of the image. In this research, specific values for $L$ and $C$ were chosen as 0.5 (corresponding to 128 on a scale of 255) and 0.0314 (equivalent to 8 on a scale of 255),

respectively. ANOVA indicated no significant difference in brightness and contrast among various groups, with p-values of 0.62 for brightness and 0.25 for contrast. Finally, we equalized the stimulus power in all SF bands (intact, R–R5). The SF power among all conditions (all SF bands, face vs. non-face and unscrambled vs. scrambled) does not vary significantly (ANOVA, p-value >0.1). SF power is calculated as the sum of the square value of the image coefficients in the Fourier domain. To create scrambled images, the original image underwent Fourier transformation, after which its phase was scrambled. Subsequently, the inverse Fourier transform was applied. Since the resulting signal may not be real, its real part was extracted. The resulting image then underwent processing through the specified filters in the primary phase.

### Task paradigm

The task was divided into two distinct phases: the selectivity phase and the main phase. Each phase comprised multiple blocks, each containing two components: the presentation of a fixation point and a stimulus. The monkey was required to maintain fixation within a window of ±1.5 degrees around the center of the monitor throughout the entire task. During the selectivity phase, there were five blocks, and stimuli were presented randomly within each block. The duration of stimulus presentation was 50 ms, while the fixation point was presented for 500 ms. The selectivity phase consisted of a total of 775 trials. A neuron was considered responsive if it exhibited a significant increase in response during the time window of 70 to 170 ms after stimulus onset, compared to a baseline window of −50 to 50 ms. This significance was determined using the Wilcoxon signed-rank test with a significance level of 0.05. Once a neuron was identified as responsive, the main phase began. In the main phase, there were 15 blocks. The main phase involved a combination of the six most responsive stimuli, selected from the selectivity phase, along with nine fixed stimuli. There was, on average, 7.54 face stimuli in each session. In each block, all stimuli were presented once in random order. The stimulus duration in the main phase was 33 ms, and the fixation point was presented for 465 ms. For the purpose of analysis, our focus was primarily on the main phase of the task.

## Neural representation

All analyses were conducted using custom code developed in Matlab (MathWorks). In total, 266 neurons (157 M1 and 109 M2) were considered for the analysis. The recorded locations, along with their SF and category selectivity are illustrated in *Appendix 1—figure 5*. Neurons were sorted using the ROSS toolbox (*Toosi et al., 2021*). In our analysis, we utilized both well-isolated single units and multi-unit activities (which represent neural activities that could not be further sorted into single units), ensuring a comprehensive representation of neural responses across the recorded population. Each stimulus in each time step was represented by a vector of $N$ elements where the $i$th element was the average response of the $i$th neuron for that stimulus in a time window of 50 ms around the given time step. We used both single- and population-level analysis. Numerous studies had examined the benefits of population representation (*Abbott and Dayan, 1999*; *Adibi et al., 2014*; *Averbeck et al., 2006*; *Dehaqani et al., 2018*). These studies have demonstrated that enhancing signal correlation within the neural data population leads to improved decoding performance for object discrimination. To maintain consistency across trials, responses were normalized using the $z$-score procedure. All time courses were based on a 50-ms sliding window with a 5-ms time step. We utilized a time window from 70 to 170 ms after stimulus onset for our analysis (except for temporal analysis). This time window was selected because the average firing rate across neurons was significantly higher than the baseline window of −50 to 50 ms (Wilcoxon signed-rank test, p-value <0.05).

## Statistical analysis

All statistical analyses were conducted as outlined in this section unless otherwise specified. In the single-level analysis, where each run involves a single neuron, pair comparisons were performed using the Wilcoxon signed-rank test, and unpaired comparisons utilized the Wilcoxon rank-sum test, both at a significance level of 0.05. The results and their SEM were reported. For population analysis, we used an empirical method, and the results were reported with their STD. To compare two paired sets of $X$ and $Y$ ($Y$ could represent the chance level), we calculated the statistic $r$ as the number of pairs where $x - y < 0$. The p-value was computed as r divided by the total number of runs, $r/M$, where $M$ is the total number of runs. When $r = 0$, we used the notation of p-value $<1/M$.

## Classification

All classifications were carried out employing the LDA method, both in population and single level. As described before, each stimulus in each block was shown by an $N$-element vector to be fed to the classifier. For face (non-face) vs. scrambled classification, only the face (non-face) and scrambled intact stimuli were used. For face vs. non-face (category) classification, only unscrambled intact stimuli were utilized. Finally, only the scrambled stimuli were fed to the classifier for the SF classification, and the labels were SF bands (R1, R2, ..., R5, multi-label classifier). In population-level analysis, averages and standard deviations were computed using a leave-p-out method, where 30% of the samples were kept as test samples in each run. All analyses were based on 1000 leave-p-out runs. To determine the onset time, one STD was added to the average accuracy value in the interval of 50 ms before to 50 ms after stimulus onset. Then, the onset time was identified as the point where the accuracy was significantly greater than this value for five consecutive time windows.

## Preferred SF

Preferred SF for a given neuron was calculated as follows:

$$PSF = \sum_i (f_{Ri} \times c_{Ri}) / \sum_i f_{Ri} \tag{4}$$

where $PSF$ is the preferred SF, $f_{Ri}$ is the average firing rate of the neuron for $Ri$, and $c_{Ri}$ is −2 for R1, −1 for R2, …, 2 for R5. When $PSF > 0$, the neuron exhibits higher firing rates for higher SF ranges on average and vice versa. To identify the number of neurons responding to a specific SF range higher than others, we performed an ANOVA analysis with a significance level of 0.05. Then, we picked the SF range with the highest firing rate for that neuron.

## SF profile

To form the SF profiles, a quadratic curve was fitted to the neuron response from R1 to R5, using exclusively scrambled stimuli. Each trial was treated as an individual sample. Neurons were categorized into three groups based on the extremum point of the fitted curve: (1) extremum is lower than R2, (2) between R2 and R4, and (3) greater than R4. Within the first group, if the neuron's response in R1 and R2 significantly exceeded (or fell below) R4 and R5, the SF profile was classified as LP (or HP). The same procedure went for the third group. Considering the second group, if the neuron response in R2 was significantly (Wilcoxon signed-rank) higher (or lower) than the response of R1 and R5, the neuron profile identified as U (or IU). Neurons not meeting any of these criteria were grouped under the flat category.

To establish sub-populations of SF/category-sorted neurons, we initially sorted the neurons according to their accuracy to decode the SF/category. Subsequently, a sliding window of size 20 was employed to select adjacent neurons in the SF- or category-sorted list. Consequently, the first sub-population comprised the initial 20 neurons exhibiting the lowest individual accuracy in decoding the SF/category. In comparison, the last sub-population encompassed the top 20 neurons with the highest accuracy in decoding SF/category.

## Separability index

The discrimination of two or more categories, as represented by the responses of the IT population, can be characterized through the utilization of the scatter matrix of category members. The scatter matrix serves as an approximate measure of covariance within a high-dimensional space. The discernibility of these categories is influenced by two key components: the scatter within a category and the scatter between categories. SI is defined as the ratio of between- to within-category scatter. The computation of SI involves three sequential steps. Initially, the center of mass for each category, referred to as $\mu$ and the overall mean across all categories, termed the total mean, $m$, was calculated. Second, we calculated the between- and within-category scatters,

$$S_i = \sum_{j,k \in C_i} (r_j - \mu_j)(r_k - \mu_k)$$

$$S_w = \sum S_i \tag{5}$$

$$S_B = \sum_{j,k \in C_i} n_i \times (\mu_i - m)(\mu_i - m)$$

where $S_i$ is the scatter matrix of the $i$th category, $r$ is the stimulus response, $S_w$ is within-category scatter, $S_B$ is the between-category scatter, and $n_i$ is the number of samples in the $i$th category. Finally, SI was computed as

$$SI = \frac{\|S_B\|}{\|S_W\|} \tag{6}$$

where $\|S\|$ indicates the norm of $S$. For additional information, please refer to the study conducted by *Dehaqani et al., 2016*.

## SNC and CMI

To examine the influence of individual neurons on population-level decoding, we introduced the concept of the SNC. It measures the reduction in decoding performance when a single neuron is removed from the population. We systematically removed each neuron from the population one at a time and measured the corresponding drop in accuracy compared to the case where all neurons were present.

To quantify the CMI between pairs of neurons, we discretized their response patterns using 10 levels of uniformly spaced bins. The CMI is calculated using the following formula:

$$CMI(n_i, n_j | c) = \sum_{n_i \in N_i, n_j \in N_j, c \in C} P(n_i, n_j, c) \times log_2 \frac{P(n_i, n_j | c)}{P(n_i | c) \times P(n_j | c)} \tag{7}$$

where $n_i$ and $n_j$ represent the discretized responses of the two neurons, and $C$ represents the conditioned variable, which can be the category (face/non-face) or the SF range (LSF (R1 and R2) and HSF (R4 and R5)). We normalized the CMI by subtracting the CMI obtained from randomly shuffled responses and added the average CMI of SF and category. CMI calculation enables us to assess the degree of information shared or exchanged between pairs of neurons, conditioned on the category or SF while accounting for the underlying probability distributions.

## Sparseness analysis

The sparseness analysis was conducted on the LDA weights, regarded as a measure of task relevance. To calculate the sparseness of the LDA weights, the neuron responses were first normalized using the $z$-score method. Then, the sparseness of the weights associated with the neurons in the LDA classifier was computed. The sparseness is computed using the following formula:

$$S = 1 - \frac{E(|w|)^2}{E(w^2)} \tag{8}$$

where $w$ is the neuron weight in LDA and $E(w^2)$ represents the mean of the squared weights of the neurons. The maximum sparseness occurs when only one neuron is active, whereas the minimum sparseness occurs when all neurons are equally active.

## Deep neural network analysis

To compare our findings with those derived from deep neural networks, we commenced by curating a diverse assortment of CNN architectures. This selection encompassed ResNet18, ResNet34, VGG11, VGG16, InceptionV3, EfficientNetb0, CORNet-S, CORTNet-RT, and CORNet-z, strategically chosen to offer a comprehensive overview of SF processing capabilities within deep learning models. Our experimentation spanned the utilization of both randomly initialized weights and pre-trained weights sourced from the ImageNet dataset. This dual approach allowed us to assess the influence of prior knowledge embedded in pre-trained weights on SF decoding. In the process of extracting feature maps, we fed our stimulus set to the models, capturing the feature maps from the last four layers,

excluding the classifier layer. Our results were primarily rooted in the final layer (preceding classification), yet they demonstrated consistency across all layers under examination. For classification and SF profiling, our methodology mirrored the procedures employed in our neural response analysis.

## Additional information

### Funding
No external funding was received for this work.

### Author contributions
Ramin Toosi, Data curation, Software, Formal analysis, Validation, Investigation, Visualization, Methodology, Writing – original draft, Writing – review and editing; Behnam Karami, Ehsan Rezayat, Conceptualization, Data curation, Methodology; Roxana Koushki, Conceptualization, Data curation, Software, Methodology; Farideh Shakerian, Conceptualization, Investigation, Methodology; Jalaledin Noroozi, Data curation, Methodology; Abdol-Hossein Vahabie, Conceptualization, Investigation; Mohammad Ali Akhaee, Conceptualization, Supervision, Validation, Project administration, Writing – review and editing; Mohammad-Reza A Dehaqani, Conceptualization, Data curation, Supervision, Validation, Investigation, Methodology, Project administration, Writing – review and editing

### Author ORCIDs
Ramin Toosi https://orcid.org/0000-0002-7099-9353
Behnam Karami https://orcid.org/0000-0001-6594-6516
Farideh Shakerian https://orcid.org/0009-0001-8426-5879
Ehsan Rezayat https://orcid.org/0000-0003-3808-1283
Abdol-Hossein Vahabie http://orcid.org/0000-0003-1603-8866
Mohammad Ali Akhaee https://orcid.org/0000-0003-3753-5618
Mohammad-Reza A Dehaqani https://orcid.org/0000-0003-4365-4365

### Ethics
The activity of neurons in the IT cortex of two male macaque monkeys weighing 10 and 11 kg, respectively, was analyzed following the National Institutes of Health Guide for the Care and Use of Laboratory Animals and the Society for Neuroscience Guidelines and Policies. The experimental procedures were approved by the Institute of Fundamental Science committee.

Reviewer #1 (Public Review): https://doi.org/10.7554/eLife.93589.4.sa1
Reviewer #2 (Public Review): https://doi.org/10.7554/eLife.93589.4.sa2
Author response https://doi.org/10.7554/eLife.93589.4.sa3

## Additional files

### Supplementary files
MDAR checklist

Source code 1. Source code for controlling stimulus presentation and juice delivery using MATLAB and the MonkeyLogic toolbox.

### Data availability
All data generated or analysed during this study are included in the manuscript and supporting files; source data files have been provided for Figures 1–6. A repository containing sample data and the code necessary to reproduce the main results is available at https://github.com/ramintoosi/spatial-frequency-selectivity (copy archived at *Toosi, 2025*).

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

## Appendix 1

### Strength of SF selectivity

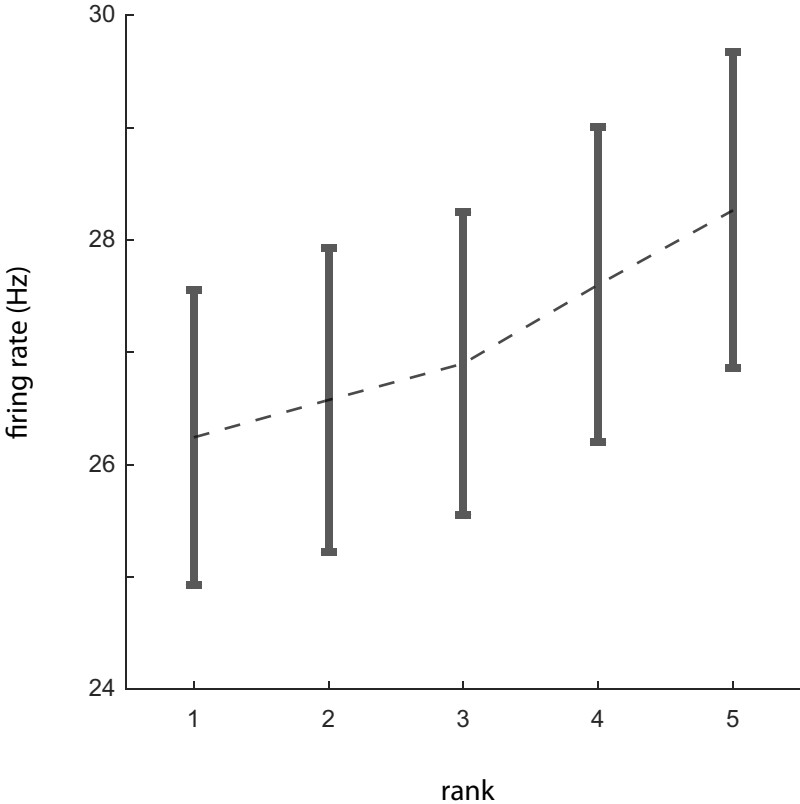

**Appendix 1—figure 1.** Strength of SF selectivity. To assess the strength of SF selectivity in IT responses, we first ranked the SF content based on the firing rate in each neuron employing half of the trials. Then, the other half is used to calculate the firing rate of each rank. Results show that the firing rate of rank 5 is significantly higher than rank 1 (p-value = $4 \times 10^{-4}$). The error bars show STD.

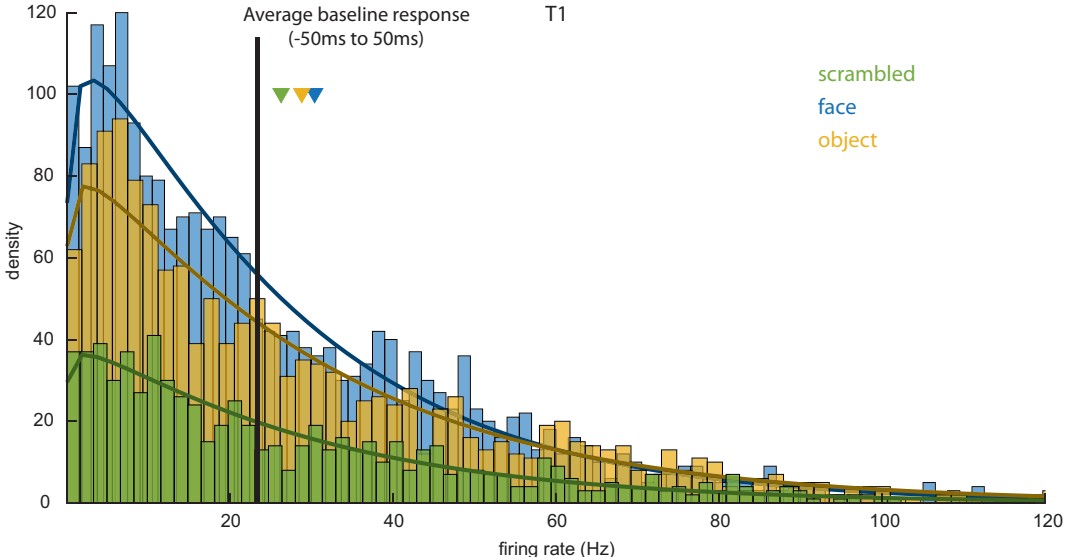

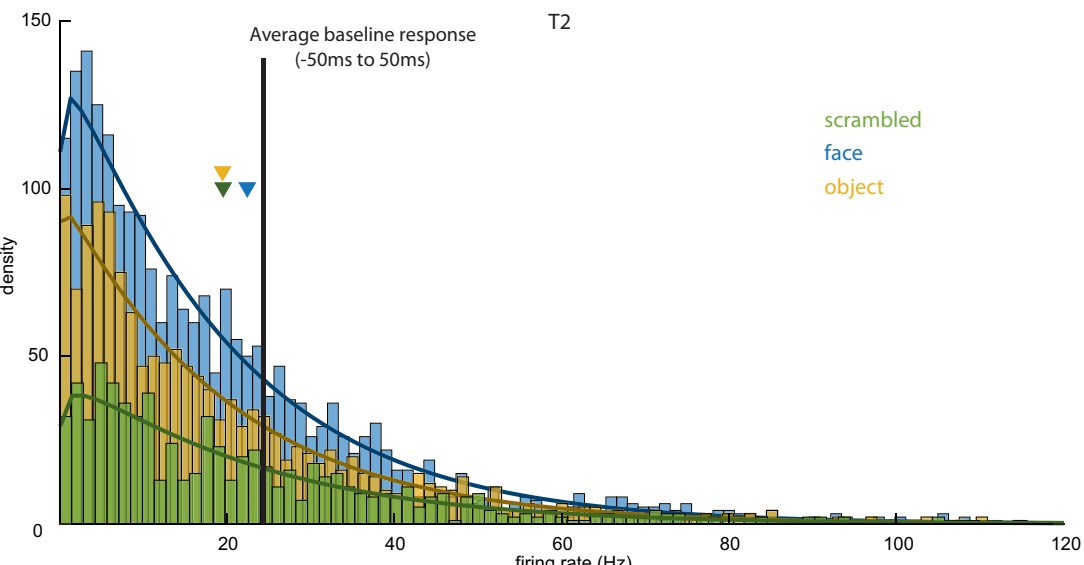

**Appendix 1—figure 2.** SF response distribution. To check the SF response strength, the histogram of IT neuron responses to scrambled, face, and non-face stimuli is illustrated in this figure. A Gamma distribution is also fitted to each histogram. To calculate the histogram, the neuron response to each unique stimulus is calculated for each neuron in spike/seconds (Hz). In the early phase, T1, the average firing rate to scrambled stimuli is 26.3 Hz which is significantly higher than the response in −50 to 50 ms which is 23.4 Hz. In comparison, the mean response to intact face stimuli is 30.5 Hz, while non-face stimuli elicit an average response of 28.8 Hz. The average net responses to the scrambled, face, and non-face stimuli are 2.9, 7.1, and 5.4 Hz, respectively. Moving to the late phase, T2, the responses to scrambled, face, and object stimuli are 19.5, 19.4, and 22.4 Hz, respectively. The corresponding average net responses are 3.9, 4.0, and 1.0 Hz below the baseline response. While the firing rates and net responses to scrambled stimuli were modest (e.g., 2.9 Hz in T1), the differences across spatial frequency (SF) bands were statistically significant ($p \approx 1e-5$) and led to a classification accuracy 24.68% above chance. This demonstrates the robustness of SF modulation in IT neurons despite low firing rates. The modest responses align with expectations for noise-like stimuli, which are less effective in driving IT neurons, yet the observed SF selectivity highlights a fundamental property of IT encoding.

## Robustness of SF profiles

To investigate the robustness of the SF profiles, considering the trial-to-trial variability, we calculated the neuron's profile using half of the trials. Then, the neuron's response to R1, R2, ..., R5 is calculated

with the remaining trials. *Appendix 1—figure 3* illustrates the average response of each profile for SF bands in each profile.

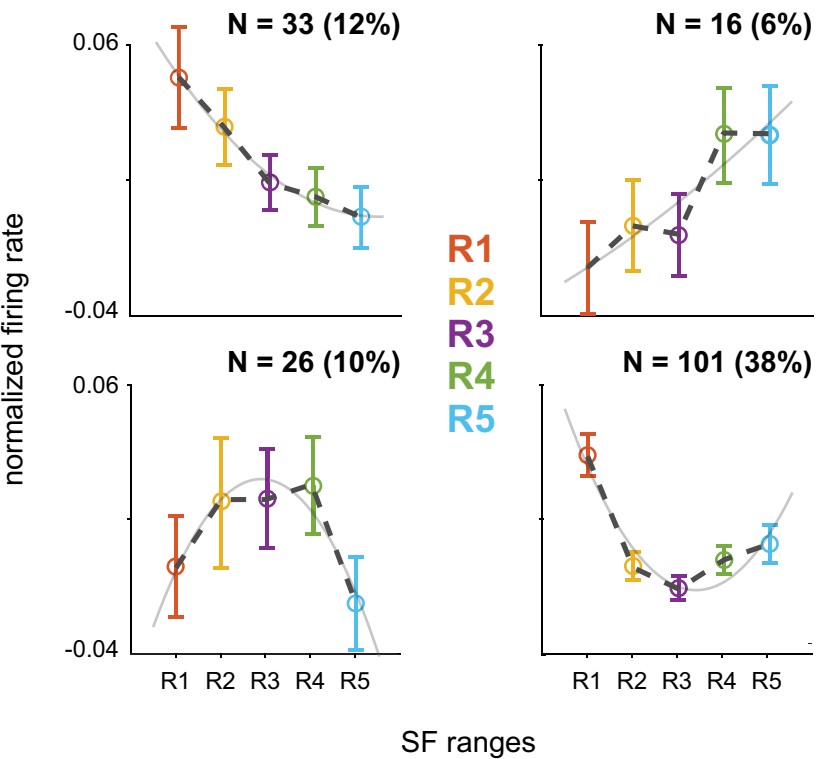

**Appendix 1—figure 3.** SF profile robustness. Profiles are calculated using half of the trials. Then, the average of the neuron responses in each profile is calculated with the remaining half. STD is illustrated with error bars.

## Extended stimulus duration supports LSF-preferred tuning

Our recorded data in the main phase contains the 200 ms version of stimulus duration for all neurons. In this experiment, we investigate the impact of stimulus duration on LSF-preferred recall and course-to-fine nature of SF decoding. As illustrated in *Appendix 1—figure 4*, the LSF-preferred nature of SF decoding (recall R1 = 0.60 ± 0.02, R2 = 0.44 ± 0.03, R3 = 0.32 ± 0.03, R4 = 0.35 ± 0.03, R5 = 0.36 ± 0.02, and R1 > R5, p-value <0.001) and course-to-fine nature of SF processing (onset times in milliseconds after stimulus onset, R1 = 87.0 ± 2.9, R2 = 86.0 ± 4.0, R3 = 93.8 ± 3.5, R4 = 96.1 ± 3.9, R5 = 96.0 ± 3.9, R1 < R5, p-value <0.001) is observed in extended stimulus duration. For the 200-ms stimulus duration, the firing rates were 27.7, 30.7, and 30.4 Hz for scrambled, face, and object stimuli in T1, respectively, and 26.2, 29.1, and 33.9 Hz in T2. The corresponding net responses were 4.3, 7.3, and 7.0 Hz in T1, and 2.8, 5.7, and 10.5 Hz in T2. While the longer stimulus duration did not substantially increase firing rates in T1, its impact was more pronounced in T2.

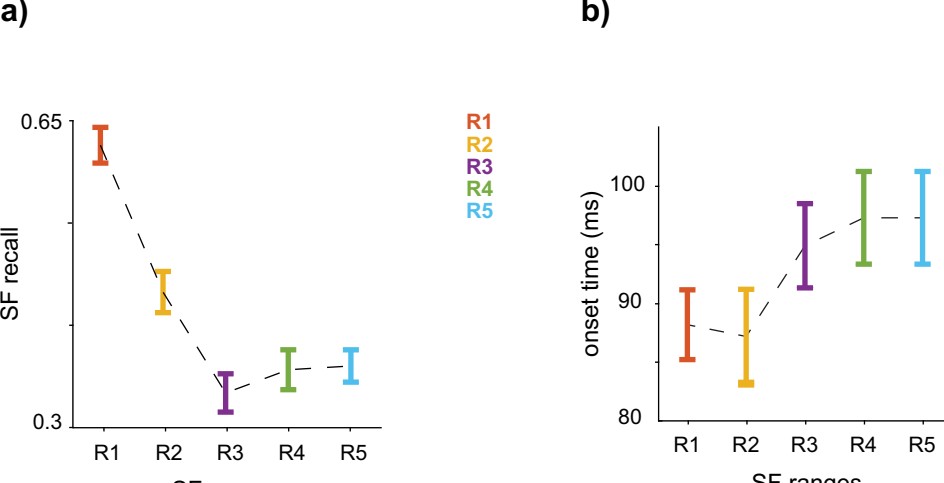

**Appendix 1—figure 4.** LSF-preferred responses with extended stimulus duration. We conducted the experiments in *Appendix 1—figure 1c* and *Appendix 1—figure 2a* with 200 ms of stimulus duration with the same method, in 70–170 ms after stimulus onset. (**a**) The recall of each SF band in the population, as elicited by scrambled stimuli and determined through the LDA method, is presented. The error bars denote the STD. The findings support the LSF-preferred nature of SF decoding observed with 33 ms of stimulus duration. (**b**) The onset time of recall for each spatial SF band in response to scrambled stimuli is depicted, with error bars representing the STD. The results imply an increasing onset time of decoding as SF values rise, as we observed in 33-ms stimulus duration.

## SF and category selectivity based on the neuron's location

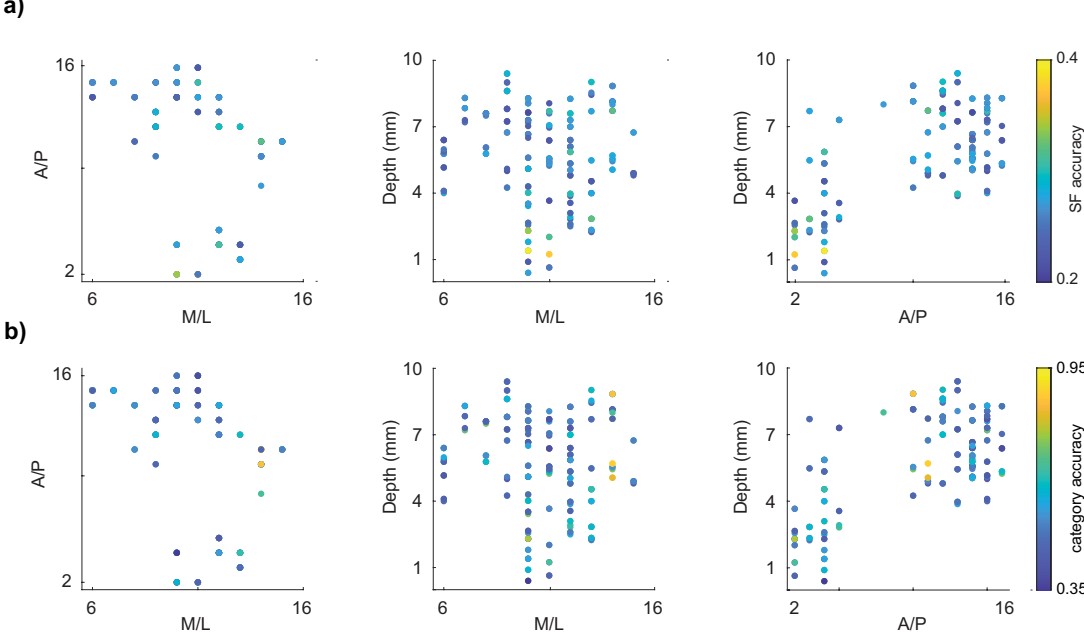

**Appendix 1—figure 5.** The SF and category selectivity of the recorded locations. The accuracy of single neurons for SF prediction (**a**) and category prediction (**b**) is illustrated for each recorded location. The x- and y-axes show anterior–posterior (A/P) or medial–lateral (M/L) hole location and the depth of the electrode in milliliters. A/P ranges from 5 mm (hole number 1) to 30 mm (hole number 18) and M/L ranges from 0 mm (hole number 1) to 23 mm (hole number 18).

## Appendix 2

### Main results for each monkey

In this section, we provide a summary of the main results for each monkey. *Appendix 1—figure 1* illustrates the key findings separately for M1 (157 neurons) and M2 (109 neurons). Regarding recall, both monkeys exhibit a decrease in recall values as the shift toward higher frequencies occurs (recall value for **M1:** R1 = 0.32 ± 0.03, R2 = 0.30 ± 0.02, R3 = 0.25 ± 0.03, R4 = 0.24 ± 0.03, and R5 = 0.24 ± 0.03. **M2:** R1 = 0.60 ± 0.03, R2 = 0.38 ± 0.03, R3 = 0.29 ± 0.03, R4 = 0.35 ± 0.03, and R5 = 0.35 ± 0.03). In both monkeys, the recall value of R1 is significantly lower than R5 (for both M1 and M2, p-value <0.001). In terms of onset, we observed a coarse-to-fine behavior in both monkeys (onset value in ms, **M1:** R1 = 84.7 ± 5.5, R2 = 82.1 ± 4.5, R3 = 90.0 ± 4.3, R4 = 86.8 ± 7.0, R5 = 103.3 ± 5.2. **M2:** R1 = 76.6 ± 1.3, R2 = 76.0 ± 1.2, R3 = 90.0 ± 4.3, R4 = 86.8 ± 2.2, R5 = 89.0 ± 1.9). Next, we examined the SF-based profiles (*Figure 3*) in M1 and M2. As depicted in *Appendix 1—figure 1c*, both monkeys exhibit similar decoding capabilities in the SF-based profiles, consistent with what we observed in *Figure 3*. In both M1 and M2, face decoding in HP significantly surpasses face/non-face decoding in all other profiles (**M1:** face SI: LP = 0.23 ± 0.05, HP = 0.91 ± 0.16, IU = 0.06 ± 0.03, U = 0.14 ± 0.02/non-face, and HP > LP, U, IU with p-value <0.001. Non-face SI: LP = 0.13 ± 0.07, HP = 0.08 ± 0.05, IU = 0.16 ± 0.09, U = 0.19 ± 0.10, and face SI in HP is greater than non-face SI in all profiles with p-value <0.001. **M2:** face SI: LP = 0.07 ± 0.03, HP = 0.38 ± 0.18, IU = 0.06 ± 0.03, U = 0.07 ± 0.05/non-face, and HP > LP, U, IU with p-value <0.001. Non-face SI: LP = 0.08 ± 0.06, HP = 0.03 ± 0.03, IU = 0.17 ± 0.04, U = 0.07 ± 0.05, and face SI in HP is greater than non-face SI in all profiles with p-value <0.001). Furthermore, in both monkeys, the non-face decoding capability in IU is significantly higher than face decoding (p-value <0.001). For neurons preferring LSF, LP profile, it is important to note the lack of consistency in responses across monkeys. This variability may reflect individual differences in neural processing or variations in sampling between subjects.

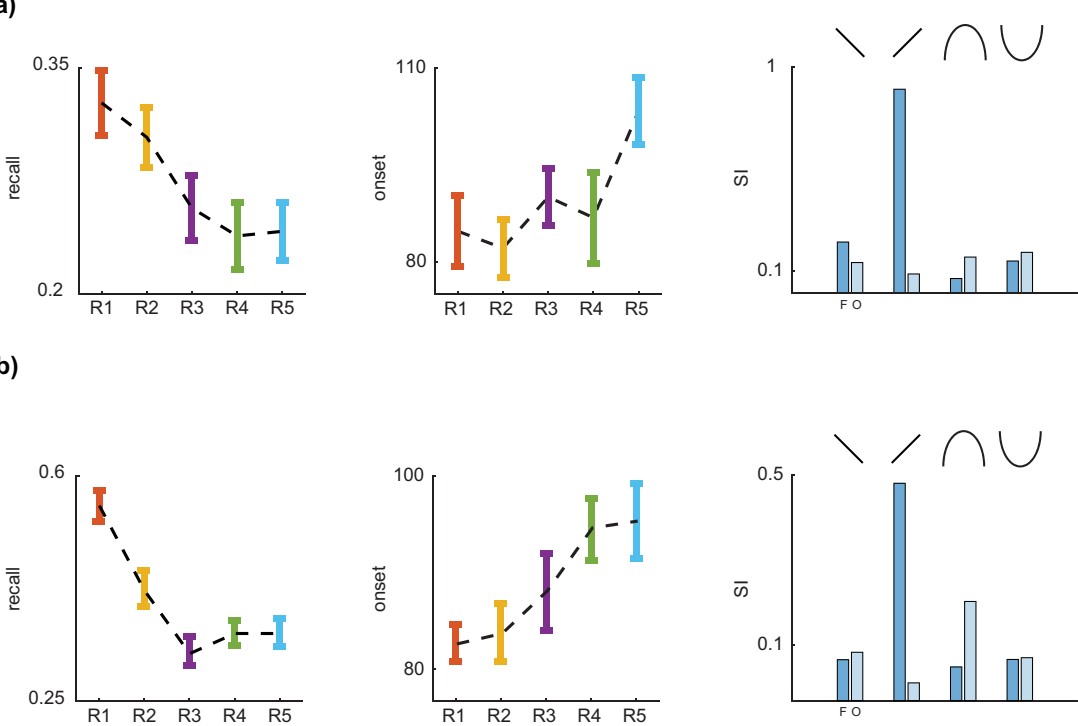

**Appendix 2—figure 1.** Main results for the two monkeys. Their call (**a**), on set of recall (**b**) and SI of each profile (**c**) is illustrated for M1 and M2, respectively. The results are consistent with our observations in the Results section. The error bars indicate STD.

