## [Editor Report · eLife Assessment]

This **useful** study aimed to examine the relationship of spatial frequency selectivity of single macaque inferotemporal (IT) neurons to category selectivity. Interesting findings in this report suggest a shift in preferred spatial frequency during the response, from low to high spatial frequencies. This agrees with a coarse-to-fine processing strategy, which is in line with multiple studies in the early visual cortex. Some of the findings were difficult to evaluate because the methods are **incomplete**. The conclusion that single-unit spatial frequency selectivity can predict object coding requires further evidence to confirm.

---

## [Referee Report · Reviewer #1 (Public Review)]

This study reports that spatial frequency representation can predict category coding in the inferior temporal cortex. The original conclusion was based on likely problematic stimulus timing (33 ms which was too brief). Now the authors claim that they also have a different set of data on the basis of longer stimulus duration (200 ms).

One big issue in the original report was that the experiments used a stimulus duration that was too brief and could have weakened the effects of high spatial frequencies and confounded the conclusions. Now the authors provided a new set of data on the basis of a longer stimulus duration and made the claim that the conclusions are unchanged. These new data and the data in the original report were collected at the same time as the authors report.

The authors may provide an explanation why they performed the same experiments using two stimulus durations and only reported one data set with the brief duration. They may also explain why they opted not to mention in the original report the existence of another data set with a different stimulus duration, which would otherwise have certainly strengthened their main conclusions.

---

## [Referee Report · Reviewer #2 (Public Review)]

Summary:

This paper aimed to examine the spatial frequency selectivity of macaque inferotemporal (IT) neurons and its relation to category selectivity. The authors suggest in the present study that some IT neurons show a sensitivity for the spatial frequency of scrambled images. Their report suggests a shift in preferred spatial frequency during the response, from low to high spatial frequencies. This agrees with a coarse-to-fine processing strategy, which is in line with multiple studies in the early visual cortex. In addition, they report that the selectivity for faces and objects, relative to scrambled stimuli, depends on the spatial frequency tuning of the neurons.

Strengths:

Previous studies using human fMRI and psychophysics studied the contribution of different spatial frequency bands to object recognition, but as pointed out by the authors little is known about the spatial frequency selectivity of single IT neurons. This study addresses this gap and shows spatial frequency selectivity in IT for scrambled stimuli that drive the neurons poorly. They related this weak spatial frequency selectivity to category selectivity, but these findings are premature given the low number of stimuli they employed to assess category selectivity.

The authors revised their manuscript and provided some clarifications regarding their experimental design and data analysis. They responded to most of my comments but I find that some issues were not fully or poorly addressed. The new data they provided confirmed my concern about low responses to their scrambled stimuli. Thus, this paper shows spatial frequency selectivity in IT for scrambled stimuli that drive the neurons poorly (see main comments below). They related this (weak) spatial frequency selectivity to category selectivity, but these findings are premature given the low number of stimuli to assess category selectivity.

---

## [Author Response]

The following is the authors’ response to the original reviews.

**Public Reviews:**

**Reviewer #1 (Public Review):**
This study reports that spatial frequency representation can predict category coding in the inferior temporal cortex.

Thank you for taking the time to review our manuscript. We greatly appreciate your valuable feedback and constructive comments, which have been instrumental in improving the quality and clarity of our work.

The original conclusion was based on likely problematic stimulus timing (33 ms which was too brief). Now the authors claim that they also have a different set of data on the basis of longer stimulus duration (200 ms).One big issue in the original report was that the experiments used a stimulus duration that was too brief and could have weakened the effects of high spatial frequencies and confounded the conclusions. Now the authors provided a new set of data on the basis of a longer stimulus duration and made the claim that the conclusions are unchanged. These new data and the data in the original report were collected at the same time as the authors report.The authors may provide an explanation why they performed the same experiments using two stimulus durations and only reported one data set with the brief duration. They may also explain why they opted not to mention in the original report the existence of another data set with a different stimulus duration, which would otherwise have certainly strengthened their main conclusions.

Thank you for your comments regarding the stimulus duration used in our experiments. We appreciate the opportunity to clarify and provide further details on our methodology and decisions.

In our original report, we focused on the early phase of the neuronal response, which is less affected by the duration of the stimulus. Observations from our data showed that certain neurons exhibited high firing rates even with the brief 33 ms stimulus duration, and the results we obtained were consistent across different durations. To avoid redundancy, we initially chose not to include the results from the 200 ms stimulus duration, as they reiterated the findings of the 33 ms duration.

However, we acknowledge that the brief stimulus duration could raise concerns regarding the robustness of our conclusions, particularly concerning the effects of high spatial frequencies. Upon reflecting on the reviewer’s comments during the first revision, we recognized the importance of addressing these potential concerns directly. Therefore, we have included the data from the 200 ms stimulus duration in our revised manuscript.

Furthermore, Our team is actively investigating the differences between fast (33 ms) and slow (200 ms) presentations in terms of SF processing. Our preliminary observations suggest similar processing of HSF in the early phase of the response for both fast and slow presentations, but different processing of HSF in the late phase. This was another reason we initially opted to publish the results from the brief stimulus duration separately, as we intended to explore the different aspects of SF processing in fast and slow presentations in subsequent studies.

I suggest the authors upload both data sets and analyzing codes, so that the claim could be easily examined by interested readers.

Thank you for your suggestion to make both data sets and the analyzing codes available for examination by interested readers.

We have created a repository that includes a sample of the dataset along with the necessary codes to output the main results. While we cannot provide the entire dataset at this time due to ongoing investigations by our team, we are committed to ensuring transparency and reproducibility. The data and code samples we have provided should enable interested readers to verify our claims and understand our analysis process.

Repository: https://github.com/ramintoosi/spatial-frequency-selectivity

**Reviewer #2 (Public Review):**
Summary:This paper aimed to examine the spatial frequency selectivity of macaque inferotemporal (IT) neurons and its relation to category selectivity. The authors suggest in the present study that some IT neurons show a sensitivity for the spatial frequency of scrambled images. Their report suggests a shift in preferred spatial frequency during the response, from low to high spatial frequencies. This agrees with a coarse-to-fine processing strategy, which is in line with multiple studies in the early visual cortex. In addition, they report that the selectivity for faces and objects, relative to scrambled stimuli, depends on the spatial frequency tuning of the neurons.Strengths:Previous studies using human fMRI and psychophysics studied the contribution of different spatial frequency bands to object recognition, but as pointed out by the authors little is known about the spatial frequency selectivity of single IT neurons. This study addresses this gap and shows spatial frequency selectivity in IT for scrambled stimuli that drive the neurons poorly. They related this weak spatial frequency selectivity to category selectivity, but these findings are premature given the low number of stimuli they employed to assess category selectivity.

Thank you for your thorough review and insightful feedback on our manuscript. We greatly appreciate your time and effort in providing valuable comments and suggestions, which have significantly contributed to enhancing the quality of our work.

The authors revised their manuscript and provided some clarifications regarding their experimental design and data analysis. They responded to most of my comments but I find that some issues were not fully or poorly addressed. The new data they provided confirmed my concern about low responses to their scrambled stimuli. Thus, this paper shows spatial frequency selectivity in IT for scrambled stimuli that drive the neurons poorly (see main comments below). They related this (weak) spatial frequency selectivity to category selectivity, but these findings are premature given the low number of stimuli to assess category selectivity.

While we acknowledge that the number of instances per condition is relatively low, the overall dataset is substantial. Specifically, our study includes a total of 180 stimuli (6 spatial frequencies × 2 scrambled/non-scrambled conditions × 15 instances, including 9 fixed and 6 non-fixed) and 5400 trials (180 stimuli × 2 durations × 15 repetitions). Conducting these trials requires approximately one hour of experimental time per session.

Extending the number of stimuli, while potentially addressing this limitation, would significantly compromise the quality of the experiment by increasing the duration and introducing potential fatigue effects in the subjects. Despite this limitation, our findings lay important groundwork by offering novel insights into object recognition through the lens of spatial frequency. We believe this work can serve as a foundation for future experiments designed to further explore and validate these theories with expanded stimulus sets.

Main points.(1) They have provided now the responses of their neurons in spikes/s and present a distribution of the raw responses in a new Figure. These data suggest that their scrambled stimuli were driving the neurons rather poorly and thus it is unclear how well their findings will generalize to more effective stimuli. Indeed, the mean net firing rate to their scrambled stimuli was very low: about 3 spikes/s. How much can one conclude when the stimuli are driving the recorded neurons that poorly? Also, the new Figure 2- Appendix 1 shows that the mean modulation by spatial frequency is about 2 spikes/s, which is a rather small modulation. Thus, the spatial frequency selectivity the authors describe in this paper is rather small compared to the stimulus selectivity one typically observes in IT (stimulus-driven modulations can be at least 20 spikes/s).

To address the concerns regarding the firing rates and the modulation of neuronal responses by spatial frequency (SF), we emphasize several key points:

(1) Significance of Firing Rate Differences: While it is true that the mean net firing rate to our scrambled stimuli was relatively low, the firing rate differences observed were statistically significant, with p-values approximately at 1e-5. This indicates that despite the low firing rates, the observed differences are reliable and unlikely to have occurred by chance.

(2) Classification Rate and Modulation by SF: Our analysis showed that the difference between various SF responses led to a classification rate of 44.68%, which is 24.68% higher than the chance level. This substantial increase above the chance level demonstrates that SF significantly modulates IT responses, even if the overall firing rates are modest.

(3) Effect Size and SF Modulation: While the effect size in terms of firing rate differences may be small, it is significant. The significant modulation of IT responses by SF, as evidenced by our statistical analyses and classification rate, supports our conclusions regarding the role of SF in driving IT responses.

(4) Expectations for Noise-like Pure SF Stimuli: We acknowledge that IT responses are typically higher for various object stimuli. Given the nature of our pure SF stimuli, which resemble noise-like patterns, we did not anticipate high responses in terms of spikes per second. The low firing rates are consistent with the expectation for such stimuli and do not undermine the significance of the observed modulation by SF.

We believe that these points collectively support the validity of our findings and the significance of SF modulation in IT responses, despite the low firing rates. We appreciate your insights and hope this clarifies our stance on the data and its implications.

We added the following description to the Appendix 1 - “Strength of SF selectivity” section:

“While the firing rates and net responses to scrambled stimuli were modest (e.g., 2.9 Hz in T1), the differences across spatial frequency (SF) bands were statistically significant (p ≈ 1e-5) and led to a classification accuracy 24.68\% above chance. This demonstrates the robustness of SF modulation in IT neurons despite low firing rates. The modest responses align with expectations for noise-like stimuli, which are less effective in driving IT neurons, yet the observed SF selectivity highlights a fundamental property of IT encoding.”

(2) Their new Figure 2-Appendix 1 does not show net firing rates (baseline-subtracted; as I requested) and thus is not very informative. Please provide distributions of net responses so that the readers can evaluate the responses to the stimuli of the recorded neurons.

We understand the reviewer’s concern about the presentation of net firing rates. In T2 (the late time interval), the average response rate falls below the baseline, resulting in negative net firing rates, which might confuse readers. To address this, we have added the net responses to the text for clarity. Additionally, we have included the average baseline response in the figure to provide a more comprehensive view of the data.

“To check the SF response strength, the histogram of IT neuron responses to scrambled, face, and non-face stimuli is illustrated in this figure. A Gamma distribution is also fitted to each histogram. To calculate the histogram, the neuron response to each unique stimulus is calculated for each neuron in spike/seconds (Hz). In the early phase, T1, the average firing rate to scrambled stimuli is 26.3 Hz which is significantly higher than the response in -50 to 50ms which is 23.4 Hz. In comparison, the mean response to intact face stimuli is 30.5 Hz, while non-face stimuli elicit an average response of 28.8 Hz. The average net responses to the scrambled, face, and non-face stimuli are 2.9 Hz, 7.1 Hz, and 5.4 Hz, respectively. Moving to the late phase, T2, the responses to scrambled, face, and object stimuli are 19.5 Hz, 19.4 Hz, and 22.4 Hz, respectively. The corresponding average net responses are 3.9 Hz, 4.0 Hz, and 1.0 Hz below the baseline response.”

(3) The poor responses might be due to the short stimulus duration. The authors report now new data using a 200 ms duration which supported their classification and latency data obtained with their brief duration. It would be very informative if the authors could also provide the mean net responses for the 200 ms durations to their stimuli. Were these responses as low as those for the brief duration? If so, the concern of generalization to effective stimuli that drive IT neurons well remains.

The firing rates for the 200 ms stimulus duration are as follows: 27.7 Hz, 30.7 Hz, and 30.4 Hz for scrambled, face, and object stimuli in T1, respectively; and 26.2 Hz, 29.1 Hz, and 33.9 Hz in T2. The average baseline firing rate (−50 to 50 ms) is 23.4 Hz. Therefore, the net responses are 4.3 Hz, 7.3 Hz, and 7.0 Hz for T1; and 2.8 Hz, 5.7 Hz, and 10.5 Hz for T2 for scrambled, face, and object stimuli, respectively.

Notably, the impact of stimulus duration is more pronounced in T2, which is consistent with the time interval of the T2 compared to T1. However, the firing rates in T1 do not show substantial changes with the longer duration. As we discussed in our response to the first comment, it is important to note that high net responses are not typically expected for scrambled or noise-like stimuli in IT neurons. Instead, the key findings of this study lie in the statistical significance of these responses and their meaningful relationship to category selectivity. These results highlight the broader implications for understanding the role of spatial frequency in object recognition.

We added the firing rates to the, Appendix 1, “Extended stimulus duration supports LSF-preferred tuning” part as follows.

“For the 200 ms stimulus duration, the firing rates were 27.7 Hz, 30.7 Hz, and 30.4 Hz for scrambled, face, and object stimuli in T1, respectively, and 26.2 Hz, 29.1 Hz, and 33.9 Hz in T2. The corresponding net responses were 4.3 Hz, 7.3 Hz, and 7.0 Hz in T1, and 2.8 Hz, 5.7 Hz, and 10.5 Hz in T2. While the longer stimulus duration did not substantially increase firing rates in T1, its impact was more pronounced in T2.”

(4) I still do not understand why the analyses of Figures 3 and 4 provide different outcomes on the relationship between spatial frequency and category selectivity. I believe they refer to this finding in the Discussion: "Our results show a direct relationship between the population's category coding capability and the SF coding capability of individual neurons. While we observed a relation between SF and category coding, we have found uncorrelated representations. Unlike category coding, SF relies more on sparse, individual neuron representations.". I believe more clarification is necessary regarding the analyses of Figures 3 and 4, and why they can show different outcomes.

Figure 3 explores the relationship between SF coding and category coding at both the single-neuron and population levels.

● Figures 3(a) and 3(b) examine the relationship between a single neuron’s response pattern and object decoding in the population.

● Figure 3(c) investigates the relationship between a single neuron’s SF decoding capabilities and object decoding in the population.

● Figure 3(d) assesses the relationship between a single neuron’s object decoding capabilities and SF decoding in the population.

In summary, Figure 3 demonstrates a relation between SF coding/response pattern at the single level and category coding at the population level.

Figure 4, on the other hand, addresses the uncorrelated nature of SF and category coding.

● Figure 4(a) shows the uncorrelated relation between a single neuron’s SF decoding capability and its object decoding capability. This suggests that a neuron's ability to decode SF does not predict its ability to decode object categories.

● Figure 4(b) illustrates that the contribution of a neuron to the population decoding of SF is uncorrelated with its contribution to the population decoding of object categories. This further supports the idea that the mechanisms behind SF coding and object coding are uncorrelated.

In summary, Figure 4 suggests that while there is a relation between SF coding and category coding as illustrated in Figure 3, the mechanisms underlying SF coding and object coding operate independently (in terms of correlation), highlighting the distinct nature of these processes.

We hope this explanation clarifies why the analyses in Figures 3 and 4 present different outcomes. Figure 3 provides insight into the relationship between SF and category coding, while Figure 4 emphasizes the uncorrelated nature of these processes. We also added the following explanation in the “Uncorrelated mechanisms for SF and category coding” section.

Based on your command, to clarify the presentation of the work, we added the following description to the “Uncorrelated mechanisms for SF and category coding” section:

“Figures 3 and 4 examine different aspects of the relationship between SF and category coding. Figure 3 highlights a relationship between SF coding at the single-neuron level and category coding at the population level. Conversely, Figure 4 demonstrates the uncorrelated mechanisms underlying SF and category coding, showing that a neuron’s ability to decode SF is not predictive of its ability to decode object categories. This distinction underscores that while SF and category coding are related at broader levels, their underlying mechanisms are independent, emphasizing the distinct processes driving each form of coding.”

(5) The authors found a higher separability for faces (versus scrambled patterns) for neurons preferring high spatial frequencies. This is consistent for the two monkeys but we are dealing here with a small amount of neurons. Only 6% of their neurons (16 neurons) belonged to this high spatial frequency group when pooling the two monkeys. Thus, although both monkeys show this effect I wonder how robust it is given the small number of neurons per monkey that belong to this spatial frequency profile. Furthermore, the higher separability for faces for the low-frequency profiles is not consistent across monkeys which should be pointed out.

We appreciate the reviewer’s concern regarding the relatively small number of neurons in the high spatial frequency group (16 neurons, 6% of the total sample across the two monkeys) and the consistency of the results. While we acknowledge this limitation, it is important to note that findings involving sparse subsets of neurons can still be meaningful. For example, Dalgleish et al. (2020) demonstrated that perception can arise from the activity of as few as ~14 neurons in the mouse cortex, supporting the sparse coding hypothesis. This underscores the potential robustness of results derived from small neuronal populations when the activity is statistically significant and functionally relevant.

Regarding the higher separability for faces among neurons preferring high spatial frequencies, the consistency of this finding across both monkeys suggests that this effect is robust within this subgroup. For neurons preferring low spatial frequencies, we agree that the lack of consistency across monkeys should be explicitly noted. These differences may reflect individual variability or differences in sampling across subjects and merit further investigation in future studies.

To address this concern, we have updated the text to explicitly discuss the small size of the high spatial frequency group, its implications, and the observed inconsistency in the low spatial frequency profiles between monkeys. We have added the following description to the discussion.

“Next, according to Figure 3(a), 6% of the neurons are HSF-preferred and their firing rate in HSF is comparable to the LSF firing rate in the LSF-preferred group. This analysis is carried out in the early phase of the response (70-170ms). While most of the neurons prefer LSF, this observation shows that there is an HSF input that excites a small group of neurons. Importantly, findings involving small neuronal populations can still be meaningful, as studies like Dalgleish et al. (2020) have demonstrated that perception can arise from the activity of as few as ~14 neurons in the mouse cortex, emphasizing the robustness of sparse coding.”

Regarding the separability of faces for the low-frequency profiles, we added the following to the appendix section,

“For neurons preferring LSF, LP profile, it is important to note the lack of consistency in responses across monkeys. This variability may reflect individual differences in neural processing or variations in sampling between subjects.”

And in the discussion:

“Our results are based on grouping the neurons of the two monkeys; however, the results remain consistent when looking at the data from individual monkeys as illustrated in Appendix 2. However, for neurons preferring LSF, we observed inconsistency across monkeys, which may reflect individual differences or sampling variability. These findings highlight the complexity of SF processing in the IT cortex and suggest the need for further research to explore these variations.”

* Henry WP Dalgleish, Lloyd E Russel, lAdam M Packer, Arnd Roth, Oliver M Gauld, Francesca Greenstreet, Emmett J Thompson, Michael Häusser (2020) How many neurons are sufficient for perception of cortical activity? eLife 9:e58889.

(6) I agree that CNNs are useful models for ventral stream processing but that is not relevant to the point I was making before regarding the comparison of the classification scores between neurons and the model. Because the number of features and trial-to-trial variability differs between neural nets and neurons, the classification scores are difficult to compare. One can compare the trends but not the raw classification scores between CNN and neurons without equating these variables.

We appreciate the reviewer’s follow-up comment and agree that differences in the number of features and trial-to-trial variability between IT neurons and CNN units make direct comparisons of raw classification scores challenging. As the reviewer suggests, it is more appropriate to focus on comparing trends rather than absolute scores when analyzing the similarities and differences between these systems. In light of this, we have revised the text to clarify that our intention was not to equate raw classification scores but to highlight the qualitative patterns and trends observed in spatial frequency encoding between IT and CNN units.

“SF representation in the artificial neural networks

We conducted a thorough analysis to compare our findings with CNNs. To assess the SF coding capabilities and trends of CNNs, we utilized popular architectures, including ResNet18, ResNet34, VGG11, VGG16, InceptionV3, EfficientNetb0, CORNet-S, CORTNet-RT, and CORNet-z, with both pre-trained on ImageNet and randomly initialized weights. Employing feature maps from the four last layers of each CNN, we trained an LDA model to classify the SF content of input images. Figure 5(a) shows the SF decoding accuracy of the CNNs on our dataset (SF decoding accuracy with random (R) and pre-trained (P) weights, ResNet18: P=0.96±0.01 / R=0.94±0.01, ResNet34 P=0.95±0.01 / R=0.86±0.01, VGG11: P=0.94±0.01 / R=0.93±0.01, VGG16: P=0.92±0.02 / R=0.90±0.02, InceptionV3: P=0.89±0.01 / R=0.67±0.03, EfficientNetb0: P=0.94±0.01 / R=0.30±0.01, CORNet-S: P=0.77±0.02 / R=0.36±0.02, CORTNet-RT: P=0.31±0.02 / R=0.33±0.02, and CORNet-z: P=0.94±0.01 / R=0.97±0.01). Except for CORNet-z, object recognition training increases the network's capacity for SF coding, with an improvement as significant as 64\% in EfficientNetb0. Furthermore, except for the CORNet family, LSF content exhibits higher recall values than HSF content, as observed in the IT cortex (p-value with random (R) and pre-trained (P) weights, ResNet18: P=0.39 / R=0.06, ResNet34 P=0.01 / R=0.01, VGG11: P=0.13 / R=0.07, VGG16: P=0.03 / R=0.05, InceptionV3: P=<0.001 / R=0.05, EfficientNetb0: P=0.07 / R=0.01). The recall values of CORNet-Z and ResNet18 are illustrated in Figure 5(b). However, while the CNNs exhibited some similarities in SF representation with the IT cortex, they did not replicate the SF-based profiles that predict neuron category selectivity. As depicted in Figure 5(c) although neurons formed similar profiles, these profiles were not associated with the category decoding performances of the neurons sharing the same profile.”

Discussion:

“Finally, we compared SF's representation trends and findings within the IT cortex and the current state-of-the-art networks in deep neural networks.”

**Recommendations for the authors:**

**Reviewer #2 (Recommendations For The Authors):**
The mean baseline firing rate of their neurons (23.4 Hz) was rather high for single IT neurons (typically around 10 spikes/s or lower). Were these well-isolated units or mainly multiunit activity?

We confirm that the recordings in our study were from both well-isolated single units and multi-unit activities (remaining after isolation neurons) sorted based on our spike sorting toolbox. The higher baseline firing rate is likely due to the experimental design, particularly the inclusion of the responsive neurons from the selectivity phase. We added the following statement to the methods section.

“In our analysis, we utilized both well-isolated single units and multi-unit activities (which represent neural activities that could not be further sorted into single units), ensuring a comprehensive representation of neural responses across the recorded population.”